



# Atmospheric deposition of organic matter at a remote site in the Central Mediterranean Sea: implications for marine ecosystem

Yuri Galletti[1], Silvia Becagli[2], Alcide di Sarra[3], Margherita Gonnelli[1], Elvira Pulido-Villena[4],

Damiano M. Sferlazzo[3], Rita Traversi[2], Stefano Vestri[1], Chiara Santinelli[1]

[1]CNR, Biophysics Institute, Pisa, Italy
[2]Department of Chemistry "Ugo Schiff", University of Florence, Italy
[3]Laboratory for Observations and Analyses of the Earth and Climate (SSPT-PROTER-OAC), ENEA, Rome, Italy
[4]Institut Méditerranéen d'Océanologie, MIO - Marseille, France

*Correspondence to:* Yuri Galletti (yuri.galletti@pi.ibf.cnr.it)

**Abstract.** Atmospheric fluxes of dissolved organic matter (DOM) were studied for the first time at the Island of Lampedusa, a remote site in the Central Mediterranean Sea (Med Sea), close to the Sahara desert, between March 19[th] 2015 and April 1[st] 2017. The main goals of this work are: to quantify total atmospheric deposition of DOM in this area and to evaluate the impact of dust deposition on DOM dynamics in the surface waters of the Mediterranean Sea. Our data show high variability in DOM deposition rates without a clear seasonality and allow to estimate a dissolved organic carbon (DOC) input from the atmosphere of 120.7 mmol DOC m$^{-2}$ y$^{-1}$. Over the entire time-series, the average dissolved organic phosphorous (DOP) and dissolved organic nitrogen (DON) contributions to the total dissolved pools were 40% and 26%, respectively. The data on atmospheric elemental ratios also show that each deposition event is characterized by a specific elemental ratio, suggesting a high variability in DOM composition and the presence of multiple sources. This study indicates that the organic substances transported by Saharan dust at Lampedusa site mainly have natural origin, especially from sea spray and that Saharan dust can be an important carrier of organic substances, even if the load of DOC associated with dust is highly variable. Our estimates suggest that atmospheric input has an impact to the Med Sea larger than to the global ocean and that DOC fluxes from the atmosphere to the Med Sea can be up to 6-fold larger than river input. Longer time series, combined with a modelling effort, are therefore mandatory in order to investigate the response of DOM dynamics in the Med Sea to the change in aerosol deposition pattern due to the effect of climate change.

## 1. Introduction

The Mediterranean Sea (Med Sea) is the largest semi-enclosed basin and one of the most oligotrophic areas in the world. It is very sensitive to natural variations in the atmosphere-ocean interactions (Mermex group, 2011). Organic matter and nutrients of natural and anthropic origin, are continuously exchanged between the ocean and the atmosphere, affecting biogeochemical cycles and the marine ecosystem. The Med Sea receives anthropogenic aerosols from the northern regions, which are characterized by the presence of important industrial sites, representing relevant sources of organic substances to the atmosphere (Guerzoni and Chester, 1996). In addition, the Sahara desert is an intermittent source of mineral dust, that can transport nutrients and organic carbon to the Basin (Goudie and Middleton, 2001; Prospero et al., 2005; Vincent et al., 2016). Atmospheric deposition of nutrients (N and P) strongly influences the marine biogeochemical cycles of the Med Sea, it has therefore received attention in the last 30 years (Migon et al., 1989; Herut et al., 2002; Ridame and Guieu, 2002; Markaki et al., 2003, 2010; Pulido-Villena et al., 2008; Djaoudi et al., 2018). Compared to inorganic nutrients, there is still very few data on the atmospheric deposition of Dissolved Organic Carbon (DOC) to the surface ocean, both at the global and local scale. Organic carbon can be removed from





the atmosphere via both wet and dry deposition (Iavorivska et al., 2016). At the global scale, wet deposition transfers about 306-580 Tg DOC yr$^{-1}$ to the surface of the Earth (Willey et al., 2000; Kanakidou et al., 2012). These values correspond to almost half of the DOC delivered to the oceans by rivers annually (IPCC, 2014). Atmospheric deposition can therefore affect regional C cycling, radiative forcing, and human health (Yan and Kim, 2012; Decina et al., 2018). In addition, the expected increase in ocean stratification due to the global warming will enhance the impact of atmospheric inputs in the surface ecosystem (Kanakidou et al., 2012). The potential magnitude of atmospheric DOC inputs to open waters and the importance of its role in the carbon cycle highlight the need for a better and robust estimation of DOC deposition.

In the last years, a few studies have reported data on atmospheric deposition of DOC to the Med Sea. Total (dry + wet) atmospheric deposition was studied in North-Western Med Sea in 2006 (Pulido-Villena et al., 2008) and in 2015 (Djaoudi et al., 2018) with contrasting results. In the first study, the highest DOC flux was observed in correspondence with a Saharan dust storm, suggesting a combination of heterogeneous reactions between organic matter and mineral dust in the troposphere. In the second study, the Saharan rain event coincided with a minimum in DOC input, suggesting the presence of an aerosol poorly enriched in organic matter (Djaoudi et al., 2018). These studies were conducted in coastal areas affected by human activities. Direct measurements of total OC (TOC) in rainwater were performed at the Crete Island (Eastern Mediterranean; Economou and Mihalopoulos, 2002). This study did not take into consideration dry deposition. None of the papers cited has studied atmospheric inputs in remote sites, far from possible pollution sources and/or large cities.

The main goals of this study are: (1) to quantify total atmospheric deposition of DOC, DON and DOP at the island of Lampedusa, representative of the remote marine environment of the central Med Sea; (2) to investigate the contribution of natural and anthropogenic sources in atmospheric DOC; (3) to estimate the impact of atmospheric deposition on marine ecosystem.

## 2. Materials and methods

### 2.1 Sampling site

Bulk atmospheric deposition (dry and wet) was collected at the Station for Climate Observations (35.52°N, 12.63°E), maintained by ENEA (the Italian National Agency for New Technologies, Energy and Sustainable Economic Development), on the island of Lampedusa, Italy (Fig. 1), (http://www.lampedusa.enea.it/).

An interesting aspect of the Med Sea is related to Dissolved Organic Matter (DOM) stoichiometry. Mediterranean DOC and Dissolved Organic Nitrogen (DON) concentrations and their ratios are similar to those reported for the global ocean (Pujo-Pay et al., 2011; Santinelli, 2015). In the surface waters (0-100 m), C:N:P ratios show that Mediterranean DOM is depleted in Dissolved Organic phosphorous (DOP). The study of C:N:P ratio of DOM in atmospheric deposition is important in order to estimate the relative contribution of atmospheric DOM input to the inventory of the surface DOM pool and to understand the fate of the three elements in the water column.

This study reports the results of analyses on deposition collected at Lampedusa island (35.52°N, 12.63°E) located in central Med Sea. Lampedusa is located in an ideal position for the study of atmospheric DOC fluxes to the open Med Sea. The site is interesting, in particular, to investigate the mineral dust contribution (mean dust deposition=7.4 g$^{m-2}$ year$^{-1}$, Vincent et., 2016) to DOC deposition. Lampedusa is a flat island far from large islands or continental areas and from relevant pollutant sources. Although influences from ship traffic emissions (Becagli et al., 2012, 2017), volcanic





aerosols (Sellitto et al., 2017), forest fires (Pace et al., 2005), and regional pollution (Pace et al., 2006), have been
documented, their contribution to the total aerosol load is small, and Lampedusa may be taken as representative for the
remote marine environment of the central Med Sea. The importance of this study area is that previous work on DOC
atmospheric deposition to the Med was essentially confined to the coastal areas, less representative of what is actually
arriving to the open Med Sea. Measurements at Lampedusa provide additional important information on the deposition
in the open Med Sea.
In addition to deposition, also measurements of $PM_{10}$ amount and chemical composition, routinely performed at
Lampedusa, are used in this study.

**2.2. Atmospheric deposition sampler**
The sampler (Fig. 1) was positioned on the roof of the ENEA climatic station located on a 45 m a.s.l. plateau on the
north-eastern coast of Lampedusa. A total of 41 samples were collected between March 19$^{th}$ 2015 and April 1$^{st}$ 2017,
every 15 days or immediately after strong rain or dust storm events. Due to logistic constraints, 9 sampling periods were
longer than 20 days. The deposition sampler is similar to those successfully employed in previous studies (Pulido-
Villena et al., 2008; Markaki et al. 2010; De Vicente et al., 2012). It is composed by a 10 L Polycarbonate bottle, with a
polyethylene funnel attached on the top; a 20 μM mesh covers the funnel stem in order to prevent contamination by
insects or organic debris. In case of wet deposition, the amount of water in the sampler was weighted then it was
collected in 250 ml polycarbonate bottles and immediately frozen. In case of dry deposition, the sampler was rinsed
with 250 mL of ultrapure MilliQ water, the sample was then collected in 250 ml polycarbonate bottles and immediately
frozen. A detailed description of sampling periods, deposition types, and collected volumes is reported in Table 1.
Samples for DOC, DON and DOP were thawed and filtered through a sterile 0.2 μm Nylon filter pre-washed with 300
ml of ultrapure water to avoid any contamination. Filtered samples were frozen until the analysis. Before the analysis,
samples were brought to room temperature (24 °C).
The concentration of soluble ions metals was measured on the samples filtered on quartz filters. These filters have low
blanks level for metals and ions respect to the determined concentration both in the soluble and particulate fraction. Just
after filtration the sample was divided in two portions, one for ionic content and the other for metal content, the latter
was spiked by 0.1 mL of sub-boiled distilled (s.b.) $HNO_3$ to preserve the metals in their soluble form. Samples was keep
refrigerate at +4°C until the analysis.

**2.3 DOC analysis**
DOC analysis were carried by a Shimadzu TOC-VCSN, equipped with a quartz combustion column filled with 1.2% Pt
on alumina pillows of ~2 mm diameter. Samples were first acidified with 2N HCl and bubbled for 3 min with $CO_2$-free
ultra-high purity air in order to remove the inorganic carbon. Replicate injections were performed until the analytical
precision was lower than 1%. A five-point linear calibration curve was determined with standard solutions of potassium
hydrogen phthalate in the same concentration range as the samples (40-400 μM). The system blank was measured every
day at the beginning and end of analyses using low-carbon Milli-Q water (<3 μM C). The instrument functioning
was assessed every day by comparison of data with DOC Consensus Reference Material (CRM), kindly provided by
Prof. D. Hansell. DOC nominal value was 41-44 μM (batch 15 Lot #07-15), DOC measured value was 42.78±1.20
(n=15) (Hansell, 2005).




**2.4 DOP and DON analysis**


Twenty-six samples out of the total 41 were analyzed for dissolved organic nitrogen (DON) and phosphorous (DOP).
The samples were collected between March 19th 2015 and November 3rd 2016.
DON was estimated by subtracting the dissolved inorganic nitrogen (DIN) from the total dissolved N (TDN). DIN and
TDN were analyzed by conventional, automated colorimetric procedure (CACP) according to Aminot and Kerouel
(2007) with an estimated limit of detection of 0.02 μM. TDN was analyzed after persulfate wet-oxidation (Pujo-Pay et
al., 1997).
DOP concentrations were determined by subtracting the inorganic form (soluble reactive phosphorus, SRP) from the
total dissolved P. SRP was measured spectrophotometrically after Murphy and Riley (1962) with a limit of detection of
0.02 μM and an analytical precision of 7% at 0.1 μM. TDP was measured as SRP after UV digestion (Armstrong et al.,
1966). The photoxidation technique included a 2 hours UV treatment in a Metrohm® 705 UV digester with a digestion
efficiency of 85 ± 3 %, assessed on a 1 μM solution of β-glycerol-phosphate.

**2.5 Ions and metals content in the deposition samples**


The concentration of soluble ions metals was measured on the samples filtered on quartz filters. These filters have low
blanks level for metals and ions respect to the determined concentration both in the soluble and particulate fraction. Just
after filtration the sample was divided in two portions, one for ionic content and the other for metal content, the latter
was spiked by 0.1 mL of sub-boiled distilled (s.b.) $HNO_3$ to preserve the metals in their soluble form.
Samples was keep refrigerate at +4°C until the analysis. Ions were determined on the solution by ion chromatography as
reported in Becagli et al. (2011).
The particulate fraction of the deposition was extracted from the quartz filter through the solubilisation procedure
reported in the EU EN14902 (2005) rule for aerosol samples. The extraction procedure was performed in a microwave
oven at 220°C for 25 min by sub-boiling distilled HNO3 and 30% ultra-pure $H_2O_2$.
Metals were determined in both soluble and particulate fractions by means of an Inductively Coupled Plasma Atomic
Emission Spectrometer (ICP-AES, Varian 720-ES) equipped with an ultrasonic nebulizer (U5000 ATC, Cetac
Technologies Inc.). Daily calibration standards (internal standard: 1 ppm Ge) were used for quantification.

**2.6 $PM_{10}$ analysis**


$PM_{10}$ (particulate matter with aerodynamic equivalent diameter lower than 10 μm) is routinely sampled on a daily basis
at the island of Lampedusa (Becagli et al., 2013; Marconi et al., 2014; Calzolai et al., 2015) by using a low-volume
dual-channel sequential sampler (HYDRA FAI Instruments) equipped with two $PM_{10}$ sampling heads, operating in
accord with UNI EN12341. The $PM_{10}$ mass was determined by weighting the Teflon filters (47 mm diameter 2 μm
nominal porosity) before and after sampling with an analytical balance in controlled conditions of temperature (20±1
°C) and relative humidity (50±5%). The estimated error on $PM_{10}$ mass is around 1% at 30 μg m⁻³ in the applied
sampling conditions. A quarter of each Teflon filter was extracted using MilliQ water (about 10 ml, accurately
evaluated by weighing) in ultrasonic bath for 15 min, and the ionic content was determined by ion chromatography as
for deposition samples (Becagli et al. 2011). Another quarter of the Teflon filter was used for the determination of
metals in the atmospheric particulate as already described for the deposition samples.






**2.7 Enrichment factor**

In order to obtain information on the DOM sources, DOM concentration is compared with concentration of Al, Na and the enrichment factor of Pb (EF Pb) in the deposition as they are marker of crustal, sea spray and anthropic source respectively.

The enrichment factor (EF) respect to crustal source for Pb, V and Ni are calculated by using Al as marker for crustal aerosol. The following equation (Eq. 1) is used for EF calculation:

$$EF\ X = \frac{(\frac{X}{Al})sample}{(\frac{X}{Al})crust} \tag{1}$$

where $(X/Al)_{sample}$ is the ratio between the metal X and Al concentrations in the sample, and $(X/Al)_{crust}$ is the same ratio in the upper continental crust as reported in Henderson and Henderson (2009). By convention, element with EF<10 are called "not enriched" having a prevailing crustal source, whereas 10<EF<100 indicate a moderate enrichment and EF>100 indicate that the element (called "enriched") has a prevailing anthropogenic source (e.g. Lai et al., 2017).

**3. Results**

**3.1 DOC atmospheric fluxes**

DOC atmospheric fluxes ranged between 0.06 and 1.78 mmol C m$^{-2}$ day$^{-1}$, with a marked variability. The sampling lasted for 746 days. The deposition was lower than 0.2 mmol DOC m$^{-2}$ d$^{-1}$ (Fig. 2 and Table 2) in half of the sampling days (52%).

In 2015, the lowest deposition rates (<0.1 C m$^{-2}$ d$^{-1}$) were measured in July (Lmp09), October (Lmp13), and November (Lmp15). The highest ones (>1.2 mmol C m$^{-2}$ d$^{-1}$) occurred between March and April (Lmp02), and in June (Lmp06), both periods were characterized by dry deposition (Fig.2 and Table 2). High DOC fluxes (>0.6 mmol C m$^{-2}$ d$^{-1}$) were also observed in March (Lmp01), May (Lmp04) and at the end of July (Lmp10), in correspondence with periods dominated by wet deposition. In 2015, the annual rainfall was 360 mm, slightly higher than the average annual rainfall at the island of Lampedusa (325 mm with 42 days of rain), (data from: http://www.arpa.sicilia.it/ and http://www.eurometeo.com/italian/climate).

In 2016, the DOC deposition rates were rather low and with a smaller variability compared to the previous year. DOC fluxes ranged between 0.1 and 0.3 mmol C m$^{-2}$ d$^{-1}$ from January to May (Lmp18 to Lmp23), and from June to August (Lmp27 to Lmp30). The highest DOC fluxes (>0.8 mmol C m$^{-2}$ d$^{-1}$) were observed in May (Lmp25) and between October and November (Lmp33; Fig. 2 and Table 2).

In 2017, DOC fluxes ranged between 0.14 and 0.92 mmol C m$^{-2}$ d$^{-1}$, from January to April (Lmp36 to Lmp41); these values are higher than those observed in the first three months of the previous year (Fig. 2 and Table 2).

Atmospheric fluxes of DOC in the wet depositions were correlated with monthly precipitation rates ($r^2$=0.47, p<0.05, n=12). The precipitation rate ranged between 2.9 and 88.5 mm.

A mean daily deposition of 0.33 mmol C m$^{-2}$ d$^{-1}$ was calculated, taking into consideration the two years (from March 2015 to April 2017), corresponding to an annual DOC flux of 120.7 mmol C m$^{-2}$ year$^{-1}$.

**3.2 DON and TDN, DOP and TDP atmospheric fluxes**



DON and Total Dissolved Nitrogen (TDN) fluxes ranged between $1.5 \cdot 10^{-3}$ and 0.25 mmol DON $m^{-2}$ $d^{-1}$ and between
$1.6 \cdot 10^{-3}$ and 0.47 mmol TDN $m^{-2}$ $d^{-1}$, respectively (Fig. 3 and Table 2). In most of the sampling period (93%), DON
deposition was lower than 0.1 mmol $m^{-2}$ $d^{-1}$. The main peaks were observed in March 2015 (Lmp01), in May (Lmp24
and Lmp25) and October 2016 (Lmp33) in correspondence with high DOC deposition (Fig. 3 and Table 2).
DOP and Total dissolved phosphorous (TDP) fluxes ranged between 0 and $2.7 \cdot 10^{-3}$ mmol DOP $m^{-2}$ $d^{-1}$ and $1 \cdot 10^{-4}$ and
$8 \cdot 10^{-5}$ mmol TDP $m^{-2}$ $d^{-1}$, respectively (Fig. 4 and Table 2). Between August 2015 and September 2016 (Lmp10-
Lmp30) both DOP and TDP showed low fluxes. In 2015, atmospheric DOP and TDP showed the highest fluxes in May
(Lmp04) and August (Lmp10). In 2016, the main peaks in DOP and TDP deposition were observed in October (Lmp31)
and November (Lmp33). The 4 peaks in atmospheric DOP and TDP (Lmp04, Lmp10, Lmp31 and Lmp33) were
responsible for 16% of total depositions and were in correspondence with high DOC fluxes (Fig. 2). It is noteworthy
that in March 2015 (Lmp01) and May 2016 (Lmp25), in correspondence with high fluxes of DOC, DON and TDP,
DOP was very low (0 and $9 \cdot 10^{-5}$ mmol $m^{-2}$ $d^{-1}$, respectively) (Table 2).
Taking into consideration the entire sampling period (March 2015 - November 2016), the mean DON and DOP daily
deposition rates were 0.032 mmol N $m^{-2}$ $d^{-1}$ and $3.8 \cdot 10^{-4}$ mmol P $m^{-2}$ $d^{-1}$, corresponding to an annual fluxes of 11.61
mmol DON $m^{-2}$ $y^{-1}$ and 0.14 mmol DOP $m^{-2}$ $y^{-1}$.
It should be noted that the these fluxes could be underestimated due to the missing samples in 2015 and 2016.

**3.3 Elemental ratios in atmospheric DOM**
DOC:DON:DOP ratios showed a marked variability in the different periods (Fig. 5 and Table 3). DOC:DON molar
ratios ranged between 2.2 (Lmp24, May 2016) and 45.9 (Lmp04, May 2015) (Fig. 5a). DOC:DOP molar ratios ranged
between 244 (Lmp10, August 2015) and 11008 (Lmp25, May 2016) (Fig. 5b). DON:DOP ratio ranged between 9.2
(Lmp10, August 2015) and 1377 (Lmp25, May 2016) (Fig. 5c). No clear seasonal cycle was observed, even if in
autumn (November 2015 and October 2016) and late spring (May 2016) depositions were very poor in P, compared to
the other two elements.

**3.4 The sources of atmospheric DOM**
Previous works indicate that soluble fractions of V and Ni in aerosol samples are specific marker of anthropic source at
Lampedusa (Becagli et al, 2012 and 2017), but in the considered samples they usually do not show enrichment factor
higher than 10, therefore their source in the deposition is mainly from crustal input.
Besides, mean values of $PM_{10}$, sea salt aerosol, dust and non-sea-salt Ca (nss Ca) mean values in $PM_{10}$ samples were
calculated over the same intervals of the deposition measurements.
In Fig. 6 we reported the DOC deposition classified on the basis of the corresponding nssCa concentration in $PM_{10}$.
Following Marconi et al. (2014), Saharan dust events are identified as those with nssCa > 950 ng/$m^3$. DOC deposition
values corresponding to average nssCa larger than the threshold (950 ng/$mg^3$) are highlighted in red. DOC deposition
corresponding to a Saharan dust event occurring in at least one day of the sampling period, are indicated in orange (Fig.
6). A detailed description of the most interesting deposition events is given below.
The mean concentration of $PM_{10}$ for Lmp01 (March 2015) was 50.1 μg $m^{-3}$, with an average dust value of 18.2 μg $m^{-3}$
(Table 4). This sample is dominated by crustal input as revealed by the values of nssCa in the aerosol (1327.6 ng/$m^3$)



and the Al concentration in the deposition (both soluble and particulate, Fig. 7). In this sample EF(Pb) indicate the low
contribution of the anthropic source. Na concentration in the deposition is 304 mg m$^{-2}$ d$^{-1}$ (Fig. 7).
Lmp02 (March-April 2015) is characterized by the second highest DOC deposition, even if no Saharan dust event
occurred in this period (Fig. 6 and 7). PM$_{10}$ mean concentration was 29 μg m$^{-3}$, the average sea-salt aerosol value was
13.6 μg m$^{-3}$ (Table 4), with a contribution to PM$_{10}$ of 47%. This sample is strongly affected by sea spray as indicated by
the Na/Al ratio that is 60-fold higher than in Lmp01, even if the concentration of Na in the deposition is slightly low
than in Lmp01.
Lmp04 (May 2015) shows a high value of DOC; during this sampling period a Saharan dust event occurred (Fig. 6), but
the concentration of Al in the deposition was quite low (Fig. 7). The PM$_{10}$ mean concentration was 26.4 μg m$^{-3}$ and the
average sea-salt in the aerosol was 8.8 g m$^{-3}$, contributing by one third to the total particulate matter. As for Lmp03 the
ratio Na/Al is quite high suggesting that sea spray dominates also in this sample.
The mean PM$_{10}$ concentration of Lmp06 (June 2015), was 23.3 μg m$^{-3}$, with an average sea-salt aerosol concentration of
13.6 μg m$^{-3}$ (Table 4). The average contribution of sea salt aerosol to the particulate matter concentration was 27%. The
peculiar characteristic of this sample is the high concentration of soluble Al and low particulate Al in the deposition
(Fig. 7). This feature is also observed in the samples Lmp10 and Lmp12 presenting quite high concentration of DOC in
July and September 2015.
Lmp25 (May 2016) was characterized by a mean PM$_{10}$ concentration of 133.7 μg m$^{-3}$ with a peak of 267.4 μg m$^{-3}$, and
with an average dust value of 42.5 μg m$^{-3}$ (Table 4). This is the highest value of PM$_{10}$ observed in the entire sampling
period and indicates the occurrence of a Saharan dust event. The average value of nssCa in the sampling days was
4815.1 ng/m$^3$, with an incredible peak of 9207 ng/m$^3$, highlighting the occurrence of an intense Saharan dust event. The
relevant Saharan dust contribution for this sample is well revealed by Al concentration (both soluble and particulate) in
the atmospheric deposition (Fig. 7).
Lmp33 (October-November 2016) and Lmp34 (November 2016) present a very indicative pattern of the two possible
source of DOC, crustal and sea spray. Lmp33 shows higher DOC concentration than Lmp34. The former is
characterized by very high Na concentration in the deposition, conversely the second is characterized by high crustal
content (as reveled by the high concentration of Al, Fig. 7).
Unfortunately PM$_{10}$ data are not available for the fourth highest DOC deposition of the entire study period (Lmp37).

**4. Discussion**
**4.1 DOC input from the atmosphere**
The relationship between monthly precipitation rates and DOC fluxes confirmed the high efficiency in DOM
atmospheric deposition of DOC via rain events in the Med Sea, as recently reported by Djaoudi et al. (2018).
Our data allowed for the quantification of the DOC annual input from the atmosphere (120.7 mmol C m$^{-2}$ y$^{-1}$); this value
is very close to that measured at Cap Ferrat peninsula (Southern France) in 2006 (129 mmol C m$^{-2}$ y$^{-1}$; Pulido-Villena et
al., 2008) and in three lakes in the western Mediterranean basin (Southern Spain, 153.3 mmol C m$^{-2}$ y$^{-1}$ in 2005; De
Vicente et al., 2012). This value is higher than that reported for the north-western Med Sea from February 2015 to July
2016, at Frioul island, Marseille Bay (59 mmol C m$^{-2}$ y$^{-1}$; Djaoudi et al., 2018). If the same sampling period is taken
into consideration for both studies (from March 2015, the beginning of sampling in Lampedusa, to July 2016, the end of
sampling at Frioul island), DOC input is 2-times higher at Lampedusa than at Frioul Island. This variability is probably



due to the different temporal and seasonal cycles of dry and wet deposition. In particular the marked differences
between these two sites could be influenced by the presence of a south-north decreasing gradient in the intensity of the
mineral dust deposition as proposed by Vincent et al. (2016). Our data also show high variability in DOC deposition
rates without a clear seasonality. If in 2015 and 2016 the highest deposition rates were between spring and autumn, in
2017 the highest deposition rates were in winter. In addition the two highest peaks observed in 2015 (Lmp02 and
Lmp06, dry deposition) together accounted for 43% of the annual DOC flux (52 mmol C $m^{-2}$ $y^{-1}$). Depending on the
origin and trajectories of the air masses, atmosphere can carry significant amounts of DOC.
Assuming that the annual DOC flux from this study (120.7 mmol C $m^{-2}$ $y^{-1}$) is valid for the whole Med Sea
(area=2.5·$10^{12}$ $m^2$), we can estimate a total input of 3.64 Tg DOC $y^{-1}$. The global estimation for wet atmospheric DOC
deposition is 306-580 Tg C $y^{-1}$ and the input to the global ocean ranges between 90 and 246 Tg C $y^{-1}$ (Willey et al.,
2000; Kanakidou et al., 2012). The global dry deposition of OC has been estimated to be 11 Tg C $y^{-1}$, (Jurado et al.,
2008) leading to a total OC deposition to the oceans of 101-247 Tg C $y^{-1}$. The comparison of these estimates indicates
that the Med Sea receives from 1.5 to 4% of the global atmospheric input of DOC, despite it covers only 0.7% of the
global oceans area.
Moreover, if we consider the riverine DOC fluxes, our values are up to 6 times larger than the estimate of the total river
input to the Med Sea (0.6-0.7 Tg DOC $y^{-1}$; Santinelli, 2015). These results confirm the lead role of atmosphere in the
transport of allocthonous DOC to the Med Sea, as suggested recently by Galletti et al. (2019).
Few episodes of Saharan outbreaks can strongly affect the annual dust flux, indeed a single outbreak can account for
40-80% of the flux (Guerzoni et al., 1997). The most intense dust deposition events in Lampedusa generally display
larger values in spring (March-June) and in autumn (Vincent et al., 2016; Bergametti et al., 1989; Loye-Pilot and
Martin, 1996; Avila et al., 1997; Ternon et al., 2010). Deposition data in this work reveal that dust events can contribute
to the annual DOC fluxes but sea spray seems the dominant source of DOC in this area.
It should also be stressed that the DOC dynamics and its annual fluxes are not only influenced by dust deposition
events. The wet deposition is also relevant, and the correlation between monthly precipitation rates and DOC fluxes
confirms the high efficiency in DOC atmospheric deposition via rain events in the Med Sea, as recently proposed by
Djaoudi et al. (2018).

### 301    4.2 Atmospheric DON, DOP input and elemental ratios

The DON annual flux (11.61 mmol N $m^{-2}$ $y^{-1}$), observed at Lampedusa, was lower than that measured at Frioul Island
(17.80 mmol N $m^{-2}$ $y^{-1}$; Djaoudi et al., 2018). Only the study by Markaki et al. (2010) reports data on atmospheric DON
fluxes and is focused on the Eastern Med Sea. These authors reported an annual flux (18.49 mmol N $m^{-2}$ $y^{-1}$) higher than
that observed at Lampedusa. The comparison among our values (0.14 mmol P $m^{-2}$ $y^{-1}$) and the few DOP data reported in
the literature shows that the fluxes at Lampedusa are markedly higher than those reported for the Western Med Sea
(0.07 mmol P $m^{-2}$ $y^{-1}$, Djaoudi et al., 2018; 0.03 mmol P $m^{-2}$ $y^{-1}$, Migon and Sandroni, 1999), whereas they are lower
than those obtained by Violaki et al. (2017) for both the West (1.16 mmol P $m^{-2}$ $y^{-1}$) and East (0.90 mmol P $m^{-2}$ $y^{-1}$) Med
Sea. Our values are instead very similar to those reported for the Eastern Med Sea in 2001 and 2002 (0.15 mmol P $m^{-2}$
$y^{-1}$) (Markaki et al., 2010).
Over the entire time-series, the average DOP and DON contributions to TDP and TDN were 40% and 26%,
respectively. These data confirm that a significant fraction of the dissolved P and N in the atmospheric deposition was





in the organic form. These values are similar to those observed in previous studies at Frioul Island (DOP 40%, DON
25%; Djaoudi et al., 2018), and in both the western and eastern Med Sea (DOP 38%; DON 32%; Markaki et al., 2010).
The similarity among the depositions collected at the two sites (Lampedusa, Central Med Sea and Frioul, North-western
Med Sea) suggests that the remote site of Lampedusa may be representative of what the Mediterranean area receives in
terms of DON and DOP, especially in the western basin.
The data on atmospheric elemental ratios show that each deposition event is characterized by a specific elemental ratio,
suggesting a high variability in DOM composition and the presence of multiple sources. Djaoudi et al (2018) observed
an average value of DOC:DON:DOP molar ratios of 1228:308:1 in atmospheric DOM, collected in the north-western
Med Sea. In the surface Med Sea, DOC:DON:DOP ratios ranges between 1050:84:1 in the western basin to 1560:120:1
in the eastern basin (Pujo-Pay et al., 2011). The average values observed in our atmospheric deposition time-series
(1909:292:1) indicate that atmospheric DOM is enriched in DOC and DON with respect to marine DOM. This
observation is also valid when we compare our values with those recently measured on marine samples collected at the
MOOSE ANTARES offshore station (north-western Med Sea) (1227:100:1, Djaoudi et al., 2018).

### 4.3 The contribution of Saharan dust to atmospheric fluxes of dissolved organic carbon

The input of Saharan dust has important effects on the chemistry of the Mediterranean aerosols and its deposition can
enrich the Med Sea with many elements (such as Co, Ni, trace metals). Very few data are available on the interactions
between organic carbon and Saharan dust, even if organic material found in the troposphere is often associated with
dust particles (Usher et al., 2003; Aymoz et al., 2004).
Our results show that Saharan dust events can represent a relevant, albeit intermittent, source of DOC to the central Med
Sea. Focusing on the different peaks of DOC deposition, our results indicate that Lmp01, Lmp04 and Lmp25 are
associated to a Saharan dust event and that the aerosol, during its route to Lampedusa, was probably enriched with
organic substances. We hypothesize that the dust particles present in the aerosol worked as condensation nuclei for
organic molecules, facilitating their accumulation and transport (Usher et al., 2003). The role of Saharan dust in the
transport of DOC is evident in Lmp25 (May 2016) characterized by high DOC, when an intense intrusion of Saharan air
masses was favored by one synoptic situation in which the role of the cyclonic circulation with a minimum depression
was significant (www.meteogiornale.it).
If all the Saharan dust deposition events (red and orange in Fig. 6) are taken into account, the input of 49.58 mmol
DOC m$^{-2}$ to Lampedusa during the study period can be estimated, this value represents ~41% of the total flux for the
entire sampling period. Instead, if only the strong dust events (red in Fig. 6) are taken into consideration a flux of 15.26
mmol DOC m$^{-2}$ can be estimated, representing 13% of the total flux. Anyhow each deposition event must be considered
individually, because it can be characterized by an enrichment of DOC or not depending on the aerosol load (Formenti
et al., 2003; Aymoz et al., 2004).
Wet deposition mainly controls the flux of Saharan dust to the Med Sea, but dry deposition can be also important
(Guerzoni et al., 1997) and its relative importance strongly depends on meteorological conditions and local emission
(Inomata et at., 2009). Some models have estimated that wet deposition represents up to 75-95% of total deposition
(Iavorivska et al., 2016). Our data confirm the importance of wet deposition, but similarly dry deposition also plays a
crucial role. Our results stress the relevance of dry deposition (32% of the total deposition during the entire sampling





period) that, in the remote site of Lampedusa, appears to be main contributor of DOC and of other chemical species, as
suggested in the past by Morales-Baquero et al. (2013).
It is also evident by our data that Saharan dust input is not always associated with high DOC input, it cannot therefore
be considered as the only process for DOC transport and deposition. For instance sample Lmp34, that shows high
concentration of dust, is not characterized by high concentration of DOC and several samples (for example Lmp02,
Lmp33 and Lmp37) characterized by high concentration of DOC, do not show high crustal content. Indeed high DOC
deposition seems often associated to sea spray transport, for instance in samples Lmp02, Lmp10, Lmp 12, Lmp 33 and
Lmp 37 (Fig. 6, in gray).
Lmp23, Lmp27, Lmp32, Lmp35 and Lmp36 were not characterized by high DOC fluxes (Fig. 6), even if these
sampling periods were characterized by at least a strong Saharan dust event (Fig. 6, in orange). This observation
supports the hypothesis that dust from Saharan region is not typically enriched with DOC, but it behaves as aggregation
center of organic molecules in the atmosphere, and depending on its route it can be enriched or not in DOC during its
route. Lmp34 further supports this hypothesis. This sample shows the third highest average nss Ca value (1092.2
ng/m$^3$), nevertheless DOC was very low (0.20 mmol m$^{-2}$ d$^{-1}$), below the daily average flux of the entire sampling period
(0.33 mmol m$^{-2}$ d$^{-1}$) (Fig. 6, Tables 2 and 4).
However the data of atmospheric Na and soluble Al suggest a very high contribution of sea spray aerosol (the highest of
the whole considered period) (Fig. 7). Therefore, for samples Lmp33 and Lmp37 the DOC source seems to be primary
marine instead of crustal as for the samples Lmp01, Lmp04, Lmp10, Lmp12 and especially Lmp25.
Lastly, it is interesting to notice that samples characterized by high values of DOC never present high EF(Pb) and the
samples presenting EF(Pb)>10 present very low DOC concentration, suggesting that anthropic sources have a small
impact on DOC deposition at Lampedusa.

### 4.4 Implications for marine ecosystem

The measurements carried out at the Island of Lampedusa clearly show that the atmosphere is an important source of
allocthonous DOC to the Central Med Sea. Very few information is available about the biological lability of
atmospheric DOC: if labile, it can be used very quickly by the microbial loop, whereas if it is mainly recalcitrant, it can
accumulate and be transported by water masses circulation.
In order to estimate the impact of atmospheric DOC deposition to the surface waters we took into consideration a mixed
layer depth (MLD) ranging between 15 and 30 m, typical of the sea close to the island of Lampedusa in September, in
according with the estimation reported by D'Ortenzio et al. (2005). Santinelli et al. (2012) observed an average DOC
concentration of 60 µM in the same area in September 1999 (in the mixed layer), and estimated a bacterial carbon
demand (BCD) of 0.32 µM C d$^{-1}$ (assuming a bacterial growth efficiency of 15%), which represents the total amount of
carbon needed to support the observed bacterial production. In September, the atmospheric DOC flux was 0.24 mmol C
m$^{-2}$ d$^{-1}$ in 2015 and 0.38 C m$^{-2}$ d$^{-1}$ in 2016, dividing them by the average MLD (22.5 m) (D'Ortenzio et al. 2005), we
estimate that the atmospheric input is expected to determine a 0.011-0.017 µM DOC d$^{-1}$ increase in the mixed layer.
Assuming that the values of BCD observed in September 1999 (0.32 µM C d$^{-1}$) are valid also for September 2015 and
2016, and that all the DOC coming from the atmosphere is labile, it could satisfy 3-5% of the daily BCD. Instead in
summer the MLD is between 10 and 15 m, with an average value of 12.5 m (D'Ortenzio et al. 2005). The DOC input
from the atmosphere is expected to increase the DOC concentration in the mixed layer by 0.008-0.079 µM C d$^{-1}$ from



June to August 2015, and by 0.013-0.014 from June to August 2016. Assuming that a BCD of 0.32 μM C d$^{-1}$ is valid
also for summer (three months) and that all the atmospheric DOC is labile, it could satisfy 3-25% of the daily BCD.
These results highlight the relevant role of atmosphere input of DOC in sustaining the bacterial productivity in the
surface layer, particularly when the column is strongly stratified.
The Mediterranean MLD seasonal variability is characterized by a basin scale deepening from November to February-
March and an abrupt stratification in April, which is maintained throughout the summer and early autumn. Even if these
data stress the potential role of atmospheric DOC in sustaining bacterial productivity in the surface ocean, a time series
of BCD, MLD and DOC concentrations in the surface layer are crucial in order to have an accurate estimation of the
DOC atmospheric input impact on the functioning of marine ecosystem. It should be also noted that a fraction of
atmospheric DOC could be recalcitrant, and therefore could be transported to depth, playing a key role in carbon
sequestration to depth. The refractory nature of a part of atmospheric DOC is hypothesized by Sánchez-Pérez et al.
(2016), who collected a 2-year time series data on Fluorescent DOM (FDOM) deposition in the North-western Med Sea
and studied the changes in the quality and quantity of marine DOM in the Barcelona coastal area (Spain). Their results
show that atmospheric inputs induced changes in the quality of organic matter, increasing the proportion of FDOM
substances in DOM pool.
Lastly, as highlighted in the previous paragraphs, the occurrence of Saharan dust events opens interesting considerations
on their impact on the marine environment. Previous studies suggested that dust inputs can promote autotrophic
production (Ridame and Guieu, 2002; Markaki et al., 2003). Instead Pulido-Villena et al. (2008) experimentally found
that heterotrophic bacteria can reduce the amount of C exported to deeper waters, because a Saharan dust event would
have induced the mineralization of 22-70% of bioavailable DOC, changing carbon sequestration.

**4. Conclusions**
Our data show that atmospheric input has a larger impact to the Med Sea than to the global ocean and DOC fluxes from
the atmosphere to the Med Sea can be up to 6-fold larger than river input.
This study indicates that the organic substances transported by Saharan dust at Lampedusa site mainly have natural
origin, especially from sea spray and that Saharan dust can be an important carrier of organic substances. The load of
DOC associated with dust is very variable and high DOC fluxes were observed also in absence of dust deposition
events.
Atmospheric C:N:P molar ratios indicate that DOM is enriched in DOC and DON with respect to marine DOM and that
the contribution of atmospheric deposition to the marine DOM stoichiometry in the Med Sea could be relevant, in
particular during the stratification period.
For future studies, atmospheric and marine DOM molar ratios (C:N:P) could be measured over time in order to obtain
information about changes in marine DOM pool. Further studies are needed to understand the link between atmospheric
inputs and marine biogeochemistry. Data on stable carbon ($\delta^{13}$C) on atmospheric DOC would be crucial in order to gain
information about its main sources. Incubation experiments should be carried out, both with aerosol rich or poor in
DOC, in order to better understand how the microbial community can respond to dust input. Lastly, longer time series,
combined with a modelling effort, are highly desirable in order to assess the response of DOM dynamics in the Med Sea
to the change in aerosol deposition pattern due to the effect of climate change.





### Author contribution

YG and CS conceived of the study and the sampling design. YG, SB, DMS collected the samples. YG, MG, SB, RT, SV analyzed the samples. YG, CS, EPV, AdS analyzed the data and all authors assisted with data discussion and contributed to the revision and editing of the final manuscript. All authors are aware of and accept responsibility for this manuscript and have approved the final submitted manuscript.

### Acknowledgements

Part of this research was supported by "Professionalità" project, funded by the *Fondazione Banca del Monte di Lombardia*. The authors thank the analytical platform PACEM (Mediterranean Institute of Oceanography) for the analysis of organic and inorganic forms of nitrogen.

*The authors declare that they have no conflict of interest.*

*The dataset generated for this study are available on request to the corresponding author.*

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





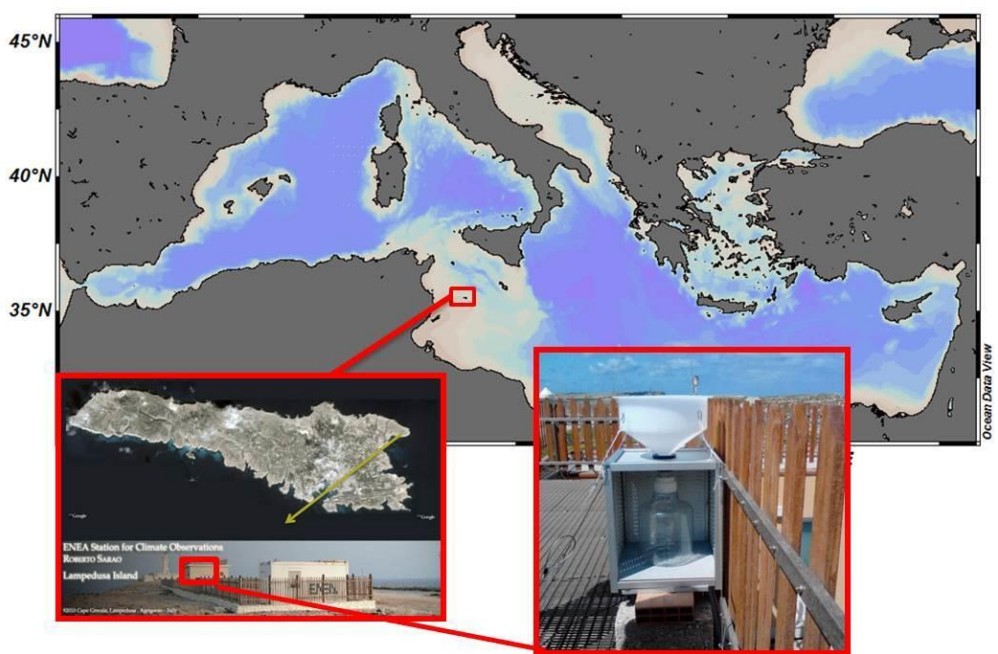

**Figure 1: Lampedusa island (35.5° N, 12.6° E) and the deposition sampler.**





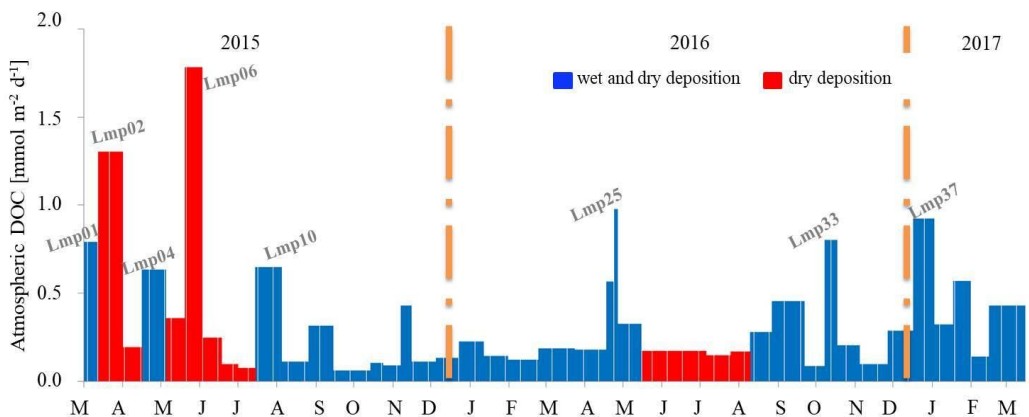

**Figure 2: Atmospheric DOC fluxes during the study period. The sign of the months is reported every 31 days.**

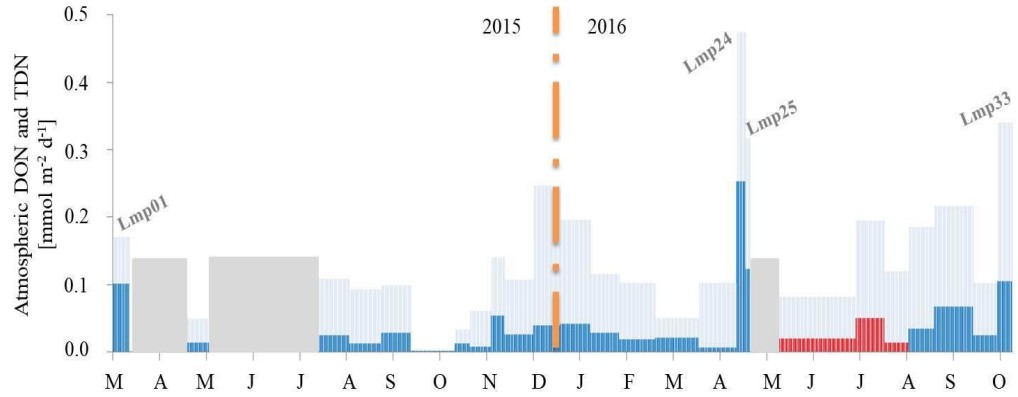

**Figure 3. Atmospheric DON in blue (wet and dry deposition) and red (dry deposition), and TDN in cyan. No data are available in the grey areas.**



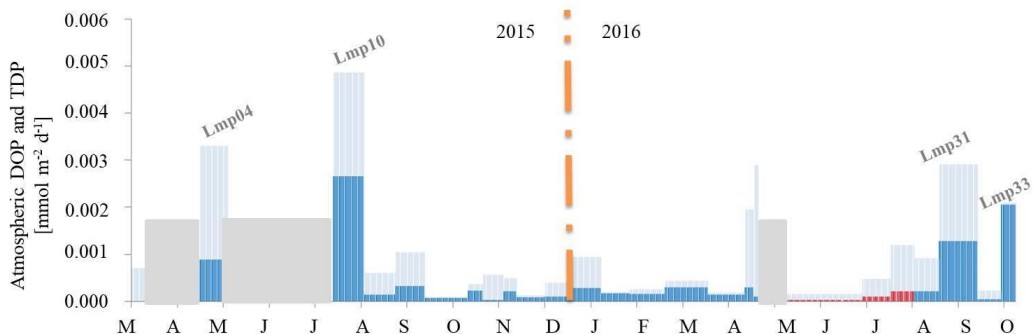

**Figure 4. Atmospheric DOP in blue (wet and dry deposition) and red (dry deposition), and TDP in cyan. No data are available in the grey areas.**



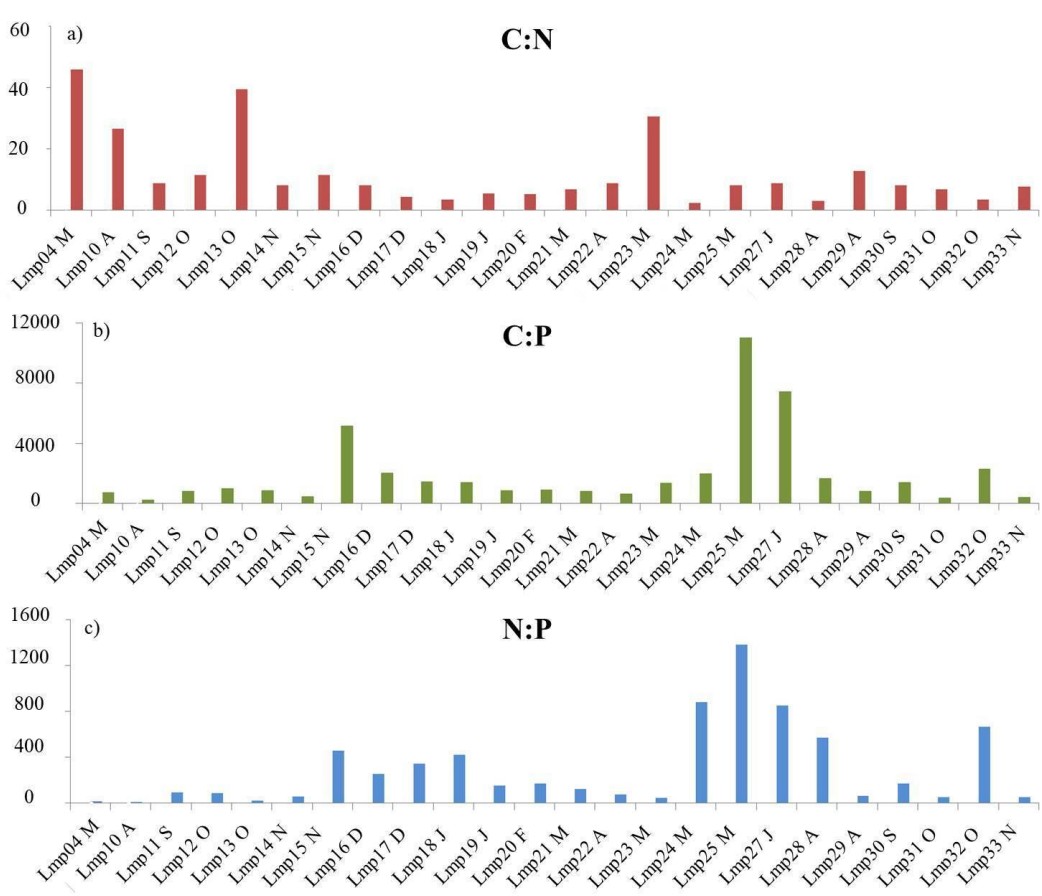

**Figure 5. Temporal evolution of C:N (a), C:P (b) and N:P (c) ratios. Sample name and the capital letter of the corresponding month of the sampling (from March, Lmp04, to November, Lmp33) are reported in the horizontal axis.**



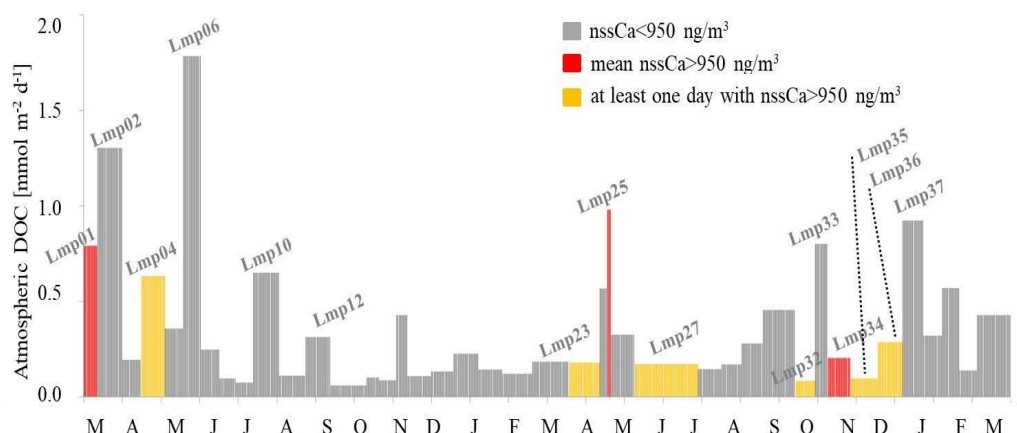

**Figure 6. Intensity of dust deposition events during the sampling period based on non-sea salt Ca (nssCa) values.**





**Figure 7. Atmospheric particulate Aluminium (a), soluble Aluminium (b), soluble Sodium (c) and enrichment**
**factor of Lead (d).**





| Sample name | Sampling period | | | Deposition type | Volume collected [L] |
|---|---|---|---|---|---|
| | Start date | End date | Total days | | |
| Lmp01 | 18/03/2015 | 28/03/2015 | 10 | wet and dry | 6 |
| Lmp02 | 28/03/2015 | 17/04/2015 | 20 | dry | 0.26 |
| Lmp03 | 17/04/2015 | 02/05/2015 | 16 | dry | 0.27 |
| Lmp04 | 02/05/2015 | 21/05/2015 | 19 | wet and dry | 1.8 |
| Lmp05 | 21/05/2015 | 05/06/2015 | 15 | dry | 0.28 |
| Lmp06 | 05/06/2015 | 19/06/2015 | 15 | dry | 0.29 |
| Lmp07 | 19/06/2015 | 04/07/2015 | 16 | dry | 0.26 |
| Lmp08 | 04/07/2015 | 17/07/2015 | 14 | dry | 0.26 |
| Lmp09 | 17/07/2015 | 31/07/2015 | 14 | dry | 0.27 |
| Lmp10 | 31/07/2015 | 21/08/2015 | 20 | wet and dry | 9 |
| Lmp11 | 21/08/2015 | 11/09/2015 | 22 | wet and dry | 2 |
| Lmp12 | 11/09/2015 | 01/10/2015 | 20 | wet and dry | 5 |
| Lmp13 | 01/10/2015 | 30/10/2015 | 29 | wet and dry | 0.5 |
| Lmp14 | 30/10/2015 | 09/11/2015 | 11 | wet and dry | 2 |
| Lmp15 | 09/11/2015 | 23/11/2015 | 14 | wet and dry | 0.6 |
| Lmp16 | 23/11/2015 | 02/12/2015 | 9 | wet and dry | 1.2 |
| Lmp17 | 02/12/2015 | 21/12/2015 | 19 | wet and dry | 1.9 |
| Lmp18 | 21/12/2015 | 08/01/2016 | 18 | wet and dry | 1.8 |
| Lmp19 | 08/01/2016 | 28/01/2016 | 20 | wet and dry | 6.1 |
| Lmp20 | 28/01/2016 | 16/02/2016 | 19 | wet and dry | 2.7 |
| Lmp21 | 16/02/2016 | 11/03/2016 | 26 | wet and dry | 2.1 |
| Lmp22 | 11/03/2016 | 09/04/2016 | 28 | wet and dry | 7.1 |
| Lmp23 | 09/04/2016 | 04/05/2016 | 26 | wet and dry | 0.3 |
| Lmp24 | 04/05/2016 | 10/05/2016 | 6 | wet and dry | 2.3 |
| Lmp25 | 10/05/2016 | 13/05/2016 | 3 | wet and dry | 1.9 |
| Lmp26 | 13/05/2016 | 01/06/2016 | 19 | wet and dry | 0.7 |
| Lmp27 | 01/06/2016 | 22/07/2016 | 50 | dry | 0.26 |
| Lmp28 | 22/07/2016 | 10/08/2016 | 19 | dry | 0.24 |
| Lmp29 | 10/08/2016 | 26/08/2016 | 16 | dry | 0.24 |
| Lmp30 | 26/08/2016 | 12/09/2016 | 17 | wet and dry | 0.8 |
| Lmp31 | 12/09/2016 | 08/10/2016 | 26 | wet and dry | 12 |
| Lmp32 | 08/10/2016 | 24/10/2016 | 16 | wet and dry | 0.5 |
| Lmp33 | 24/10/2016 | 03/11/2016 | 10 | wet and dry | 11 |
| Lmp34 | 03/11/2016 | 21/11/2016 | 18 | wet and dry | 12 |
| Lmp35 | 21/11/2016 | 13/12/2016 | 22 | wet and dry | 1.7 |
| Lmp36 | 13/12/2016 | 02/01/2017 | 20 | wet and dry | 9.5 |
| Lmp37 | 02/01/2017 | 19/01/2017 | 17 | wet and dry | 6.5 |
| Lmp38 | 19/01/2017 | 03/02/2017 | 15 | wet and dry | 1.5 |
| Lmp39 | 03/02/2017 | 17/02/2017 | 14 | wet and dry | 5 |
| Lmp40 | 17/02/2017 | 03/03/2017 | 14 | wet and dry | 0.75 |
| Lmp41 | 03/03/2017 | 01/04/2017 | 29 | wet and dry | 5.5 |





**Table 1. Sampling period, type of deposition and volume for the 41 samples collected at the Island of**
**Lampedusa.**



| Sample name | DOC fluxes [mmol m$^{-2}$ d$^{-1}$] | DON fluxes [mmol m$^{-2}$ d$^{-1}$] | TDN fluxes [mmol m$^{-2}$ d$^{-1}$] | DOP fluxes [mmol m$^{-2}$ d$^{-1}$] | TDP fluxes [mmol m$^{-2}$ d$^{-1}$] |
|---|---|---|---|---|---|
| Lmp01 | 0.80 | 0.10 | 0.17 | 0 | $7 \cdot 10^{-4}$ |
| Lmp02 | 1.30 | n.a. | n.a. | n.a. | n.a. |
| Lmp03 | 0.19 | n.a. | n.a. | n.a. | n.a. |
| Lmp04 | 0.63 | 0.01 | 0.05 | $9 \cdot 10^{-4}$ | $3 \cdot 10^{-3}$ |
| Lmp05 | 0.36 | n.a. | n.a. | n.a. | n.a. |
| Lmp06 | 1.78 | n.a. | n.a. | n.a. | n.a. |
| Lmp07 | 0.25 | n.a. | n.a. | n.a. | n.a. |
| Lmp08 | 0.10 | n.a. | n.a. | n.a. | n.a. |
| Lmp09 | 0.07 | n.a. | n.a. | n.a. | n.a. |
| Lmp10 | 0.65 | 0.02 | 0.11 | $3 \cdot 10^{-3}$ | $5 \cdot 10^{-3}$ |
| Lmp11 | 0.11 | 0.01 | 0.09 | $1 \cdot 10^{-4}$ | $6 \cdot 10^{-4}$ |
| Lmp12 | 0.31 | 0.03 | 0.10 | $3 \cdot 10^{-4}$ | $1 \cdot 10^{-3}$ |
| Lmp13 | 0.06 | $1.5 \cdot 10^{-3}$ | $1.6 \cdot 10^{-3}$ | $7 \cdot 10^{-5}$ | $8 \cdot 10^{-5}$ |
| Lmp14 | 0.10 | 0.01 | 0.03 | $2 \cdot 10^{-4}$ | $4 \cdot 10^{-4}$ |
| Lmp15 | 0.09 | $8 \cdot 10^{-3}$ | 0.06 | $2 \cdot 10^{-5}$ | $6 \cdot 10^{-4}$ |
| Lmp16 | 0.43 | 0.05 | 0.14 | $2 \cdot 10^{-4}$ | $5 \cdot 10^{-4}$ |
| Lmp17 | 0.11 | 0.03 | 0.11 | $7 \cdot 10^{-5}$ | $1 \cdot 10^{-4}$ |
| Lmp18 | 0.13 | 0.04 | 0.25 | $9 \cdot 10^{-5}$ | $4 \cdot 10^{-4}$ |
| Lmp19 | 0.23 | 0.04 | 0.20 | $3 \cdot 10^{-4}$ | $9 \cdot 10^{-4}$ |
| Lmp20 | 0.14 | 0.03 | 0.12 | $2 \cdot 10^{-4}$ | $2 \cdot 10^{-4}$ |
| Lmp21 | 0.12 | 0.02 | 0.10 | $2 \cdot 10^{-4}$ | $3 \cdot 10^{-4}$ |
| Lmp22 | 0.18 | 0.02 | 0.05 | $3 \cdot 10^{-4}$ | $4 \cdot 10^{-4}$ |
| Lmp23 | 0.18 | $6 \cdot 10^{-3}$ | 0.10 | $1 \cdot 10^{-4}$ | $2 \cdot 10^{-4}$ |
| Lmp24 | 0.57 | 0.25 | 0.47 | $3 \cdot 10^{-4}$ | $2 \cdot 10^{-3}$ |
| Lmp25 | 0.98 | 0.12 | 0.32 | $9 \cdot 10^{-5}$ | $3 \cdot 10^{-3}$ |
| Lmp26 | 0.33 | n.a. | n.a. | n.a. | n.a. |
| Lmp27 | 0.17 | 0.02 | 0.08 | $2 \cdot 10^{-5}$ | $1 \cdot 10^{-4}$ |
| Lmp28 | 0.14 | 0.05 | 0.19 | $9 \cdot 10^{-5}$ | $5 \cdot 10^{-4}$ |
| Lmp29 | 0.17 | 0.01 | 0.12 | $2 \cdot 10^{-4}$ | $1 \cdot 10^{-3}$ |
| Lmp30 | 0.28 | 0.04 | 0.18 | $2 \cdot 10^{-4}$ | $9 \cdot 10^{-4}$ |
| Lmp31 | 0.45 | 0.07 | 0.22 | $1 \cdot 10^{-3}$ | $3 \cdot 10^{-3}$ |
| Lmp32 | 0.08 | 0.02 | 0.10 | $4 \cdot 10^{-5}$ | $2 \cdot 10^{-4}$ |
| Lmp33 | 0.80 | 0.10 | 0.34 | $2 \cdot 10^{-3}$ | $2 \cdot 10^{-3}$ |
| Lmp34 | 0.20 | n.a. | n.a. | n.a. | n.a. |
| Lmp35 | 0.10 | n.a. | n.a. | n.a. | n.a. |
| Lmp36 | 0.29 | n.a. | n.a. | n.a. | n.a. |
| Lmp37 | 0.92 | n.a. | n.a. | n.a. | n.a. |
| Lmp38 | 0.32 | n.a. | n.a. | n.a. | n.a. |
| Lmp39 | 0.57 | n.a. | n.a. | n.a. | n.a. |
| Lmp40 | 0.14 | n.a. | n.a. | n.a. | n.a. |
| Lmp41 | 0.43 | n.a. | n.a. | n.a. | n.a. |

**Table 2. Atmospheric fluxes of DOC, DON, TDN, DOP and TDP at the Island of Lampedusa.**






| Sample | Sampling date | C:N | C:P | N:P |
|---|---|---|---|---|
| Lmp01 | 28/03/2015 | 7.78 | n.a. | n.a. |
| Lmp04 | 21/05/2015 | 45.87 | 715.08 | 15.59 |
| Lmp10 | 21/08/2015 | 26.57 | 244.38 | 9.20 |
| Lmp11 | 11/09/2015 | 8.67 | 807.94 | 93.15 |
| Lmp12 | 01/10/2015 | 11.37 | 977.79 | 85.98 |
| Lmp13 | 30/10/2015 | 39.44 | 864.07 | 21.91 |
| Lmp14 | 09/11/2015 | 8.02 | 449.04 | 56.00 |
| Lmp15 | 23/11/2015 | 11.26 | 5131.65 | 455.83 |
| Lmp16 | 02/12/2015 | 7.97 | 2036.66 | 255.42 |
| Lmp17 | 21/12/2015 | 4.24 | 1448.37 | 341.90 |
| Lmp18 | 08/01/2016 | 3.34 | 1406.60 | 420.55 |
| Lmp19 | 28/01/2016 | 5.38 | 832.69 | 154.79 |
| Lmp20 | 16/02/2016 | 5.09 | 882.80 | 173.40 |
| Lmp21 | 11/03/2016 | 6.63 | 812.40 | 122.55 |
| Lmp22 | 09/04/2016 | 8.78 | 645.65 | 73.53 |
| Lmp23 | 04/05/2016 | 30.48 | 1353.57 | 44.41 |
| Lmp24 | 10/05/2016 | 2.24 | 1976.03 | 882.33 |
| Lmp25 | 13/05/2016 | 7.99 | 11008.94 | 1377.41 |
| Lmp27 | 22/07/2016 | 8.73 | 7405.29 | 848.62 |
| Lmp28 | 10/08/2016 | 2.89 | 1641.49 | 568.76 |
| Lmp29 | 26/08/2016 | 12.66 | 796.68 | 62.95 |
| Lmp30 | 12/09/2016 | 8.06 | 1376.27 | 170.77 |
| Lmp31 | 08/10/2016 | 6.74 | 356.03 | 52.84 |
| Lmp32 | 24/10/2016 | 3.41 | 2275.72 | 666.53 |
| Lmp33 | 03/11/2016 | 7.68 | 389.57 | 50.73 |

**Table 3. C:N:P molar ratios in atmospheric DOM.**





| Sample name | Mean PM$_{10}$ [µg/m$^3$] | Mean sea salt aerosol [µg/m$^3$] | Mean dust [µg/m$^3$] | Mean nssCa [ng/m$^3$] |
|---|---|---|---|---|
| Lmp01 | 50.1 | 13.0 | 18.2 | 1327.6 |
| Lmp02 | 29.0 | 13.6 | n.a. | 62.2 |
| Lmp03 | 28.1 | 9.8 | n.a. | 371.6 |
| Lmp04 | 26.4 | 8.8 | 4 | 351.7 |
| Lmp05 | 16.7 | 4.6 | n.a. | 87.3 |
| Lmp06 | 23.1 | 6.1 | n.a. | 166.1 |
| Lmp07 | 22.2 | 7.1 | n.a. | 139.3 |
| Lmp08 | 26.5 | 5.4 | n.a. | 311.6 |
| Lmp09 | 28.3 | 8.0 | n.a. | 188.2 |
| Lmp10 | 29.1 | 5.2 | 3.4 | 492.7 |
| Lmp11 | n.a. | n.a. | n.a. | n.a. |
| Lmp12 | n.a. | n.a. | n.a. | n.a. |
| Lmp13 | n.a. | n.a. | n.a. | n.a. |
| Lmp14 | n.a. | n.a. | n.a. | n.a. |
| Lmp15 | n.a. | n.a. | n.a. | n.a. |
| Lmp16 | n.a. | n.a. | n.a. | n.a. |
| Lmp17 | n.a. | n.a. | n.a. | n.a. |
| Lmp18 | n.a. | n.a. | n.a. | n.a. |
| Lmp19 | n.a. | n.a. | n.a. | n.a. |
| Lmp20 | n.a. | n.a. | n.a. | n.a. |
| Lmp21 | n.a. | n.a. | n.a. | n.a. |
| Lmp22 | n.a. | n.a. | n.a. | n.a. |
| Lmp23 | 39.5 | 18.3 | 3.8 | 488.1 |
| Lmp24 | 30.7 | 18.7 | 1.2 | 154 |
| Lmp25 | 133.7 | 15.5 | 42.5 | 4815.1 |
| Lmp26 | 25.9 | 13.1 | 1.5 | 168.8 |
| Lmp27 | 26.2 | 9.3 | 2.3 | 319.5 |
| Lmp28 | 24.7 | 8.8 | 1.8 | 161.5 |
| Lmp29 | 25 | 9.6 | 1.0 | 235.9 |
| Lmp30 | 22.4 | 5.1 | n.a. | 330.8 |
| Lmp31 | 24.5 | 5.6 | n.a. | 286.2 |
| Lmp32 | 32.9 | 8.7 | n.a. | 772.5 |
| Lmp33 | 31.8 | 11.8 | n.a. | 344.2 |
| Lmp34 | 35.3 | 7.8 | n.a. | 1092.2 |
| Lmp35 | 22.3 | 7.5 | 0.4 | 394 |
| Lmp36 | 35.8 | 12.3 | 4.6 | 661.5 |
| Lmp37 | n.a. | n.a. | n.a. | n.a. |
| Lmp38 | n.a. | n.a. | n.a. | n.a. |
| Lmp39 | n.a. | n.a. | n.a. | n.a. |
| Lmp40 | n.a. | n.a. | n.a. | n.a. |





| Lmp41 | n.a. | n.a. | n.a. | n.a. |

**Table 4. The PM$_{10}$, sea salt aerosol, dust and non-sea salt Ca (nss Ca) mean values of the atmospheric DOC**
**deposition.**