# Peer review of "Atmospheric deposition of organic matter at a remote site in the Central Mediterranean Sea: implications for marine ecosystem"

_Biogeosciences, 2020_

## Referee Comment (RC1) · Anonymous Referee #1 · 24 Feb 2020

The manuscript addresses the atmospheric deposition of organic matter in the Mediterranean, for which there is little data available. It quantifies such deposition in the small island of Lampedusa in the Central Mediterranean, in terms of carbon, nitrogen and phosphorus. It also tries to untangle possible sources of such organic matter. In this aspect the manuscript is less conclusive as there is no good relationship to aerosol origin or type of deposition. The conclusion is that the OM is mainly coming from sea-spray that the different air masses pick up and transport to wind up depositing. It could be in large part but really it is just a hypothesis that needs further exploration. Also, I was surprised not to consider wind direction properties when analyzing deposited material. Lampedusa is a small island but I would not be surprised that when wind blows

from directions other than due East, and especially when it blows over the island from the West, substantial OM could be picked up from the island itself. A third aspect of the manuscript deals with estimating the local and Mediterranean basin-wide importance of such deposition estimates for the biogeochemical functioning of the Mediterranean. I like this part myself but I have to admit it is the least elaborated since it is based on assumptions that will be hardly met. For instance, calculations based on the extension to the whole Mediterranean of the measured OM deposition at Lampedusa. Given it is so variable and without a clear reason, I would expect variability to increase when other locations are taken into account. Also, the lability of the deposited organic matter is an unknown, so the final role of the marine biota is also unknown. But anyhow, I like these exercises.

Thus, to me the main value of the manuscript is to provide a much needed data series of OM deposition measurements. The methods are standard within the field and thus assure quality control. Maybe I am not clear whether monthly data were calculated and how or whether just sample data was provided always? or in what cases? That is, how where data treated when more than 1 sample per month was available? How was the data split when covering periods from two consecutive months?, etc. I understand that sample data is clearly reported in Fig. 5, but how were the rest treated is a bit mysterious, especially since bars have unequal width within and between figures.

In line 150 it is also important to know the flow rate of the low-volume sampler. Also, I guess that because of physical flow rate constraints a 1 $\mu$m filter could not be used. That would have been much more desirable since there tend to be organic rich particles at the very fine particle ranges, and they would have been missed, not a minor issue in this paper on OM. I would like the authors to comment on the choice of a 2 $\mu$m filter to collect particles.

The manuscript is well structured and balanced. The title is informative of the contents. The language is proficient. Figures should be uniformized or clarified in aspects such as the x-axis but are otherwise well done.

Other than that, I have no major concerns publishing the manuscript pretty much as it is.

———————————————————

---

## Referee Comment (RC2) · Anonymous Referee #2 · 20 Mar 2020

This paper presents a sound dataset concerning the dry and wet deposition fluxes of dissolved organic matter sampled for 2 years and a half at the island of Lampedusa (Italy). This site, in the central Mediterranean, is appropriately taken to represent the interaction atmosphere-sea surface in a remote marine environment. It is a well written paper which addresses a topic of interest: the role of DOM (and its components DON and DOP) deposition in the western Mediterranean. It explores the role of the frequent Saharan intrusions, a very interesting point since few studies have dealt with the interactions between organic carbon and Saharan dust. Finally, it specifically addresses the role of this atmospheric deposition for marine productivity. The quantification of N and P atmospheric deposition to the Mediterranean has been previously addressed in

many papers, the most relevant of them are adequately cited by the authors. However, Id like to bring to the authors attention the work of Izquierdo et al. 2012 in Atmospheric Environment. Atmospheric phosphorus deposition in a near-coastal rural site in the NE Iberian Peninsula and its role in marine productivity, since it will provide more data for comparison, discussion and understanding of the role of African sources in marine biogeochemistry, and the relative contribution of dry and wet deposition. The layout of the paper and data treatment are OK, and I have only a few suggestions, which I list below. Introduction Lines 34-35. Industrial pollution can also be originated from North Africa as has been shown in the work of Rodríguez et al (2011). Transport of desert dust mixed with North African industrial pollutants in the subtropical Saharan Air Layer. Atmospheric Chemistry and Physics 11, 6663–6685. I think it is worth considering.

Line 40. The work of Izquierdo et al 2012 could be included in this list of references, since it deals with how P dep influences the marine biogeochemical cycle in the western Med Sea.

Line 45. This sentence should be revised as it is not true that atmospheric deposition affects radiative forcing and human health. Aerosols in the atmosphere do, but not deposition.

Material and methods

I recommend to make some reorganization of the text, since some paragraphs in this section in fact correspond better to the Introduction. E.g. the paragraph dealing with the explanation of the Mediterranean seawater DOM stoichiometry compared to the world oceans (lines 70-75) should be moved to the Introduction. Same thing with the paragraph justifying the appropriateness of Lampedusa as representing an unpolluted site in the central Med.

Line 78. Revise the notation of units of mean dust deposition Line 95. polycarbonate, not in capital letter Paragraph 104-108. Please list in this text the ions and metals analyzed Line 105 and 136. blank levels, instead of blanks level Lines 135-140. This

has been already exposed in lines 104-108. Line 144. I see that the particulates retained in the filters (after wet and dry deposition filtration) was analysed. But the procedure of digestion and analysis is not reported. Same thing for particulates from the PM10 samples (line157). This should be described in the M&M.

Results Line 202. Here there is an error, since the upper limit of TDP is 5*10 exp-3 (as deduced from Table 2). Line 244. Error in unit: 8.8 ug m-3

Discussion In this section I'd like a more in deep discussion of dry versus wet deposition and its relation to meteorology.

High DOC deposition was recorded in Lmp25 (May 2016) and also in Lmp1 (end of March 2015) and Lmp 4 (May2015) coinciding with Saharan dust but low DOC was found in Saharan events during autumn and winter. In view of this clear seasonal differentiation, one could hypothesize that there is a role of pollen attached to desert particles in these spring events (end March-May) and this pollen would contribute DOC. This process would not occur in the other seasons (winter and autumn of no pollen production). This is a possible explanation that needs further attention. However, there are some reports in the literature of joint pollen and dust transport: for example, Van Campo and Quet (1982) identified pollen types transported from North Africa to south France together with mineral desert dust, Franzen et al. 1994 documented the arrival of pollen from the Mediterranean to Fennoscandia during a dust event. Pollen originating in Morocco was detected South Spain (Cabezudo et al. 1997) and various pollen types (Cannabis, Cupressus, Pinus, Platanus and Sambucus) were observed in Cordoba (South Spain) exclusively during dust African events (Cariñanos et al. 2004).

Figures In fig 2, 3 and 4, include a legend to indicate the color of wet and dry deposition.

---

## Author Comment (AC1) · 15 Apr 2020

(REFEREE) The manuscript addresses the atmospheric deposition of organic matter in the Mediterranean, for which there is little data available. It quantifizes such deposition in the small island of Lampedusa in the Central Mediterranean, in terms of carbon, nitrogen and phosphorus. It also tries to untangle possible sources of such organic matter. In this aspect the manuscript is less conclusive as there is no good relationship to aerosol origin or type of deposition. The conclusion is that the OM is mainly coming from sea spray that the different air masses pick up and transport to wind up depositing. It could be in large part but really it is just a hypothesis that needs further exploration.

[Figure]

(AUTHORS) We really thank the reviewer for his/her appreciation of our work. We totally agree that regarding the possible sources of organic matter, we just reported some hypotheses. With the available data-set in fact we were just able to make an hypothesis about the main sources of DOM in the different sampling periods. The sampling periods cover a generally wide time interval, and the deposition data are the result of integrating deposition over air masses of different origin and with different aerosol characteristics and loads. This makes very difficult to find a correlation between aerosol origin and DOC input, since the different sources are mixed in our samples. If we would have reduced the sampling periods, we would have had less variability in the sources, but in most of them we would have not had enough DOM to do all the analysis. Two weeks was therefore the best compromise we were able to find. We will add in the revised manuscript a sentence highlighting that in order to understand the link between aerosol origin and DOM concentration and quality further exploration is mandatory.

(REFEREE) Also, I was surprised not to consider wind direction properties when analyzing deposited material. Lampedusa is a small island but I would not be surprised that when wind blows from directions other than due East, and especially when it blows over the island from the West, substantial OM could be picked up from the island itself.

(AUTHORS) We thank the reviewer for this comment. At first, we took into consideration the air masses trajectories, but then we realized that there are some limitations in the use of wind direction to infer the aerosol sources. Firstly, as above reported, the sampling periods cover a generally wide time interval, and the deposition data are the result of integrating deposition over air masses of different origin and with different aerosol characteristics and loads. Thus, in some cases information on the wind might suggest what are the dominant (if existing) wind conditions during a specific period; but due to possible differences in aerosol amounts and deposition, these conditions may not be representative of the integrated samples. Secondly, the use of wind direction and speed identify the air mass origin (which is not what the reviewer is

suggesting) may be problematic, since trajectories arriving to Lampedusa may take different paths depending on the synoptic conditions. A more elaborated approach would be required (e.g., trajectory reconstruction using wind, as in Becagli et al., 2012; or modeled backward trajectories based on meteorological analyses, as in Marconi et al., 2014). However, also in this case, the relatively long duration of the sampling interval and the variability of the deposition would prevent a robust attribution of the source regions. Wind measurements conversely, as correctly suggested by the reviewer, might potentially provide useful information on the impact of local sources. The main local source area of anthropogenic particles is in the sectors between South and South-East of the sampling site; in this sectors there are the Lampedusa town, the power plant, the airport and the port. Previous studies have shown that wind from these directions is relatively infrequent, and the impact of these sectors is estimated to be negligible (see e.g., Artuso et al., 2009, with respect to atmospheric $CO_2$ measurements; Calzolai et al., 2015, with respect to PM10 measurements). This impact is expected to be even lower over samples integrated over are relatively large number of days. Due to these reasons, we have preferred to associate the deposition samples' characteristics with those of PM10 samples collected daily and over the same time period. This allows to use aerosol properties measured in the same time intervals and influenced by the same sources to infer some overall conditions. Previous studies (e.g., Becagli et al., 2012, 2013, 2017; Marconi et al., 2014; Calzolai et al, 2016) have been dedicated at linking the PM10 measured composition with different aerosol sources. These ideas will be discussed in the revised manuscript and a sentence about the main wind affecting our sampling site will be added in the materials and methods section.

(REFEREE) A third aspect of the manuscript deals with estimating the local and Mediterranean basin-wide importance of such deposition estimates for the biogeochemical functioning of the Mediterranean. I like this part myself but I have to admit it is the least elaborated since it is based on assumptions that will be hardly met. For instance, calculations based on the extension to the whole Mediterranean of the measured OM deposition at Lampedusa. Given it is so variable and without a clear reason,

I would expect variability to increase when other locations are taken into account. Also, the lability of the deposited organic matter is an unknown, so the fi̧nal role of the marine biota is also unknown. But anyhow, I like these exercises. Thus, to me the main value of the manuscript is to provide a much needed data series of OM deposition measurements.

(AUTHORS) We totally agree with the reviewer and we are aware that reporting the calculations based on the extension of our results to the entire Med Sea is a risk, because it is nor simple, nor appropriate, to assume that what is observed at one location is valid for the entire basin. However, we consider this as a first conceptual exercise that uses the new results from our study to give an estimate of the implications of DOM deposition for marine ecosystem, that needs to be supported by additional data. To the best of the author's knowledge only one paper reports that a not-well quantified fraction of atmospheric DOM can be recalcitrant. Due to the lack of information, we decided to discuss implications taking into consideration both the possibilities: DOM is labile and DOM is recalcitrant. In the revised manuscript we will better stress the need for further investigations about the biological lability of DOM coming from the atmosphere In order to better stress that these calculations are a conceptual exercise, we can add the following sentences in the revised manuscript: "A conceptual exercise can be made in order to give an estimate of the implications of DOM deposition for marine ecosystem." "Even if we are aware that these assumptions are hardly meet, in particular the estimate of DOC input to the whole Med Sea, based on the data collected in Lampedusa, we think that these calculations can give an idea of the relevant role that atmosphere input of DOC can have in sustaining the bacterial productivity in the surface layer, in particular when the water column is strongly stratified."

(REFEREE) The methods are standard within the fi̧eld and thus assure quality control. Maybe I am not clear whether monthly data were calculated and how or whether just sample data was provided always? or in what cases? That is, how where data treated when more than 1 sample per month was available? How was the data split

when covering periods from two consecutive months?, etc.

(AUTHORS) We apologize for the inaccuracy. We did not calculate monthly data, we reported the sample data. The width of the bars in the figures 2, 3, 4, 6 and 7 refers to the duration of the sampling period (Table 1). This aspect will be clarified in the revised manuscript. As a general rule, samples were collected every ∼15 days, or immediately after strong rain or dust storm events. However, due to logistic problems the sampling period was longer than 20 days for 9 depositions (Table 1). The DOC, DON and DOP fluxes, reported in the text and in the figures 2, 3, 4, 6 and 7 were calculated using the following formula: XFlux =X·V/A·d where X is the concentration of DOC, DON or DOP measured in the sample and expressed in $\mu$M; V is the volume of rain collected by the sampler (expressed in L) or the volume of Milli-Q water used to wash the funnel walls in case of dry deposition (250 ml); A is the area of the funnel (0.1018 m2), and d refers to the number of days of the sampling period. The DOC, DON and DOP fluxes are reported in the figures considering the flux corresponding to each sampling period. A paragraph with this explanation can be added in the Materials and methods section of the revised manuscript in order to clarify these calculations.

(REFEREE) I understand that sample data is clearly reported in Fig. 5, but how were the rest treated is a bit mysterious, especially since bars have unequal width within and between figures.

(AUTHORS) The bars in the figure 5 corresponded to the C:N:P molar ratios (see Table 3), so they referred to a number, not a flux, and this is the reason why the width of the bars is always the same in figure 5, in contrats to fig. 2, 3, 4, 6 and 7, where the width of the bars is different since it refers to the length of sampling periods. However, thanks to the reviewer comment, we realized that making all these figures with the same format is misunderstanding. We will therefore redo the figure 5 without bar but using a symbol and we will clarify in the text how the figures are made.

(REFEREE) In line 150 it is also important to know the flow rate of the low-volume

sampler. Also, I guess that because of physical flow rate constraints a 1 $\mu$m filter could not be used. That would have been much more desirable since there tend to be organic rich particles at the very fine particle ranges, and they would have been missed, not a minor issue in this paper on OM. I would like the authors to comment on the choice of a 2 $\mu$m filter to collect particles.

(AUTHORS) The filters used in this study are those usually used for aerosol sampling, they have a nominal porosity of 2 $\mu$m, but they are certified for 99% efficiency for particles having 0.3 $\mu$m diameter. The sampling flow is maintained constant at 2.3 m3/h in order to maintain constant the sampling heads cut-off (10$\mu$m) as reported in the European rule UNI EN12341. In order to clarify these concepts for a broad number of readers the text can be changed as follows: "PM10 (particulate matter with aerodynamic equivalent diameter lower than 10 $\mu$m) is routinely sampled on a daily basis at the island of Lampedusa (Becagli et al., 2013; Marconi et al., 2014; Calzolai et al., 2015) by using a low-volume dual-channel sequential sampler (HYDRA FAI Instruments) equipped with two PM10 sampling heads, operating at constant flow of 2.3 m3/h in accord with the European rules for aerosol monitoring (UNI EN12341). Aerosol is collected on 47 mm diameter Teflon filters (PALL Gelman) having 2 $\mu$m nominal porosity but certified to have 99% retention efficiency for 0.3 $\mu$m diameter particles. The PM10 mass was determined by weighting the Teflon filters before and after sampling with an analytical balance in controlled conditions of temperature (20$\pm$1 °C) and relative humidity (50$\pm$5%)."

(REFEREE) The manuscript is well structured and balanced. The title is informative of the contents. The language is proficient. Figures should be uniformed or clarified in aspects such as the x-axis but are otherwise well done. Other than that, I have no major concerns publishing the manuscript pretty much as it is. (AUTHORS) We really thank the reviewer for his/her appreciation of our manuscript. As above reported, we will rework the figures in order to uniform them and to eliminate any misunderstanding.

[Figure]

Please also note the supplement to this comment:
https://www.biogeosciences-discuss.net/bg-2020-14/bg-2020-14-AC1-supplement.pdf
* * *

---

## Author Comment (AC2) · 15 Apr 2020

(REFEREE) This paper presents a sound dataset concerning the dry and wet deposi-tion fluxes of dissolved organic matter sampled for 2 years and a half at the island of Lampedusa (Italy). This site, in the central Mediterranean, is appropriately taken to represent the interaction atmosphere-sea surface in a remote marine environment. It is a well written paper which addresses a topic of interest: the role of DOM (and its components DON and DOP) deposition in the western Mediterranean. It explores the role of the frequent Saharan intrusions, a very interesting point since few studies have dealt with the interactions between organic carbon and Saharan dust. Finally, it

specifically addresses the role of this atmospheric deposition for marine productivity. The quantification of N and P atmospheric deposition to the Mediterranean has been previously addressed in many papers, the most relevant of them are adequately cited by the authors. However, I'd like to bring to the authors attention the work of Izquierdo et al. 2012 in Atmospheric Environment. Atmospheric phosphorus deposition in a near-coastal rural site in the NE Iberian Peninsula and its role in marine productivity, since it will provide more data for comparison, discussion and understanding of the role of African sources in marine biogeochemistry, and the relative contribution of dry and wet deposition. The layout of the paper and data treatment are OK, and I have only a few suggestions, which I list below.

(AUTHORS) We really thank the reviewer for his/her appreciation of our manuscript. In the revised version, all the comments and suggestions will be taken into consideration. In the revised manuscript, the work of Izquierdo et al. (2012) will be cited and discussed as suggested by the reviewer.

(REFEREE) Introduction Lines 34-35. Industrial pollution can also be originated from North Africa as has been shown in the work of Rodríguez et al (2011). Transport of desert dust mixed with North African industrial pollutants in the subtropical Saharan Air Layer. Atmospheric Chemistry and Physics 11, 6663–6685. I think it is worth considering.

(AUTHORS) In the introduction of the revised manuscript, we will add this reference and a sentence about the possible contribution of pollution from North Africa.

(REFEREE) Line 40. The work of Izquierdo et al 2012 could be included in this list of references, since it deals with how P dep influences the marine biogeochemical cycle in the western Med Sea.

(AUTHORS) In the introduction of the revised manuscript, we will add this reference.

(REFEREE) Line 45. This sentence should be revised as it is not true that atmospheric

deposition affects radiative forcing and human health. Aerosols in the atmosphere do, but not deposition.

(AUTHORS) We agree with the reviewer that the sentence was not clear. In the revised manuscript, we propose to change it as follows: "Atmospheric deposition of organic carbon can therefore affect regional C cycling (Yan and Kim, 2012; Decina et al., 2018)."

(REFEREE) Material and methods I recommend to make some reorganization of the text, since some paragraphs in this section in fact correspond better to the Introduction. E.g. the paragraph dealing with the explanation of the Mediterranean seawater DOM stoichiometry compared to the world oceans (lines 70-75) should be moved to the Introduction.

(AUTHORS) We apologize for this inaccuracy, in the revised manuscript, this sentences will be moved to the Introduction.

(REFEREE) Same thing with the paragraph justifying the appropriateness of Lampedusa as representing an unpolluted site in the central Med.

(AUTHORS) In agreement with this comment, in the revised manuscript we will add a subsection at the end of the introduction explaining why we choose Lampedusa Island for this work.

(REFEREE) Line 78. Revise the notation of units of mean dust deposition

(AUTHORS) OK.

(REFEREE) Line 95. polycarbonate, not in capital letter Paragraph

(AUTHORS) OK.

(REFEREE) 104-108. Please list in this text the ions and metals analyzed

(AUTHORS) We will list the ions and metals in section 2.5 and we will deleted this part since it is also reported at lines 135-140, as noted by the reviewer.

(REFEREE) Line 105 and 136. blank levels, instead of blanks level

(AUTHORS) OK.

(REFEREE) Lines 135-140. This has been already exposed in lines 104-108.

(AUTHORS) As above reported, we will delete the lines 104-108 and we will add here the list of metals and ions.

(REFEREE) Line 144. I see that the particulates retained in the filters (after wet and dry deposition filtration) was analysed. But the procedure of digestion and analysis is not reported. Same thing for particulates from the PM10 samples (line157). This should be described in the M&M.

(AUTHORS) In agreement with this comment, in the revised manuscript, the description of the procedure will be added in the Material and Methods section.

(REFEREE) Results Line 202. Here there is an error, since the upper limit of TDP is 5*10 exp-3 (as deduced from Table 2).

(AUTHORS) We apologize for the inaccuracy. The mistake will be corrected in the revised manuscript.

(REFEREE) Line 244. Error in unit: 8.8 ug m-3

(AUTHORS) We apologize for the inaccuracy. The mistake will be corrected in the revised manuscript.

(REFEREE) Discussion In this section I'd like a more in deep discussion of dry versus wet deposition and its relation to meteorology.

(AUTHORS) In agreement with this comment, in the revised manuscript, we will add some information on dry versus wet deposition and its relation to meteorology. In particular, in the results we will report the annual rainfall during 2016 and we will add the following sentence: "Precipitation shows a significant interannual variability and is

concentrated in autumn and winter, with a maximum in October. Intense precipitation events, which are relatively infrequent, are generally associated with frontal passages and winds from the Northern sectors. Very dry conditions characterize late spring and summer." In the discussion (paragraph 4.2), we will add the following sentence : "All of the analyzed samples, except few cases in summer 2016, are relative to dry+wet conditions. Although the DON and DOP recorded during the dry samplings are generally on the low end side of the measured range (see Table 2), no information on the role played by wet and dry deposition processes may be drawn at this stage, due to the limited number of dry samples." Regarding DOC input, in the discussion (paragraph 4.1) we reported that "It should also be stressed that the DOC dynamics and its annual fluxes are not only influenced by dust deposition events. The wet deposition is also relevant, and the correlation between monthly precipitation rates and DOC fluxes confirms the high efficiency in DOC atmospheric deposition via rain events in the Med Sea, as recently proposed by Djaoudi et al. (2018)." In the literature, the wet atmospheric deposition is considered the main pathway for the removal of organic carbon from the atmosphere. Our data show that dry deposition is also important and we have reported a detailed discussion of this point in the paragraph 4.3:" Some models have estimated that wet deposition represents up to 75-95% of total deposition (Iavorivska et al., 2016). Our data confirm the importance of wet deposition, but similarly dry deposition also plays a crucial role. Our results stress the relevance of dry deposition (32% of the total deposition during the entire sampling period) that, in the remote site of Lampedusa, appears to be main contributor of DOC and of other chemical species, as suggested in the past by Morales-Baquero et al. (2013)." In the revised manuscript, a sentence about the need of further studies aimed at clarifying the relationship between atmospheric deposition and meteorology will be added in the conclusions.

(REFEREE) High DOC deposition was recorded in Lmp25 (May 2016) and also in Lmp1 (end of March 2015) and Lmp 4 (May2015) coinciding with Saharan dust but low DOC was found in Saharan events during autumn and winter. In view of this clear seasonal differentiation, one could hypothesize that there is a role of pollen attached

to desert particles in these spring events ( end March-May) and this pollen would contribute DOC. This process would not occur in the other seasons (winter and autumn of no pollen production). This is a possible explanation that needs further attention. However, there are some reports in the literature of joint pollen and dust transport: for example, Van Campo and Quet (1982) identified pollen types transported from North Africa to south France together with mineral desert dust, Franzen et al. 1994 documented the arrival of pollen from the Mediterranean to Fennoscandia during a dust event. Pollen originating in Morocco was detected South Spain (Cabezudo et al. 1997) and various pollen types (Cannabis, Cupressus, Pinus, Platanus and Sambucus) were observed in Cordoba (South Spain) exclusively during dust African events (Cariñanos et al. 2004).

(AUTHORS) We really thank the reviewer for this interesting suggestion. The contribution of pollen to atmospheric DOC in spring is an interesting hypothesis to test. In the revised manuscript, the following sentence will be added in the discussion (paragraph 4.3): "In addition Lmp01 (end of March 2015), Lmp04 (May 2015) and Lmp25 (May 2016) show a seasonality that could be linked to the transport of pollen attached to desert particles in the spring events, and this pollen would contribute to atmospheric DOC input in spring (end of March- May). Pollen originating in Morocco was detected in South Spain (Cabezudo et al., 1997) and various pollen types (Cannabis, Cupressus, Pinus, Platanus and Sambucus) were observed in Cordoba (South Spain) exclusively during dust African events (Cariñanos et al., 2004). This process would not occur in the other seasons (winter and autumn), when no pollen production occurs." A sentence about the need of further studies will be also added in the conclusions.

(REFEREE) Figures In fig 2, 3 and 4, include a legend to indicate the color of wet and dry deposition.

(AUTHORS) In the revised manuscript, a legend will be included in figures 2, 3 and 4.

Please also note the supplement to this comment:

https://www.biogeosciences-discuss.net/bg-2020-14/bg-2020-14-AC2-supplement.pdf

---

## Author Response (AR1)

[revised manuscript text omitted]

**Reply to Editor**

*(EDITOR)*

*One of major conclusion of this manuscript is the organic species deposited in Lampedusa coming from natural sources (sea spray and dust), notably in summer. However, your discussion is supported by a too limited literature about OM sources in atmospheric aerosols in this area. One of the reviewers reminded you that you did not consider the potential role played by pollen.*

(AUTHORS)

We thank the editor for this comment and we apologize for the missing references. In the revised manuscript, all the suggested paper about OM sources in atmospheric aerosols are cited and we stressed the potential role played by pollen. The following sentences were added in the discussion (paragraph 4.3): . "*The role of secondary organic aerosols as a source of organic matter in the Mediterranean Sea is well documented (Arndt et al., 2017; Michoud et al., 2017; Rinaldi et al., 2017) and could be relevant at Lampedusa.*" and *"In addition Lmp01 (end of March 2015), Lmp04 (May 2015) and Lmp25 (May 2016) show a seasonality that could be linked to the transport of pollen attached to desert particles in the spring events, and this pollen would contribute to atmospheric DOC input in spring (end of March- May). Pollen originating in Morocco was detected in South Spain (Cabezudo et al., 1997) and various pollen types (Cannabis, Cupressus, Pinus, Platanus and Sambucus) were observed in Cordoba (South Spain) exclusively during dust African events (Cariñanos et al., 2004). This process would not occur in the other seasons (winter and autumn), when no pollen production occurs."* A sentence about the need of further studies was also added in the conclusions.

*It is also surprising that your discussion does not take into account the presence of atmospheric secondary organic aerosols as a source of organic matter at Lampedusa in spring and summer (see Mallet et al., ACP, 2019) but generally in Med Sea (Arndt et al., 2017; Michoud et al., 2017; Rinaldi et al., 2017).*

We thank the editor for this comment and we apologize for the inaccuracy. The following sentence was added in the discussion (paragraph 4.1): "*The role of secondary organic aerosols as a source of organic matter in the Mediterranean Sea is well documented (Arndt et al., 2017; Michoud et al., 2017; Rinaldi et al., 2017) and could be relevant at Lampedusa.*"

*Moreover, it is known that the regional pollution at Lampedusa is higher than presented in your manuscript (e.g. Pace et al. (2006) have shown that clean marine aerosol conditions are rare at*

*Lampedusa, contrary to your sentence p3L81) and a discussion on air masses trajectories is available in*
*Mallet et al., 2019. So please add this information in your corrected version before publication.*

In agreement with this comment, the sentence at P3 L81 was reworked as follows: *"Although the*
*island of Lampedusa is a remote marine environment of the central Med Sea, influences from ship traffic*
*emissions (Becagli et al., 2012, 2017), volcanic aerosols (Sellitto et al., 2017), forest fires (Pace et al.,*
*2005), and regional pollution (Pace et al., 2006), have been documented".*

**Reply to Referee 1**

*(REFEREE)*
*The manuscript addresses the atmospheric deposition of organic matter in the Mediterranean, for*
*which there is little data available. It quantifies such deposition in the small island of Lampedusa in the*
*Central Mediterranean, in terms of carbon, nitrogen and phosphorus. It also tries to untangle possible*
*sources of such organic matter. In this aspect the manuscript is less conclusive as there is no good*
*relationship to aerosol origin or type of deposition. The conclusion is that the OM is mainly coming from*
*sea spray that the different air masses pick up and transport to wind up depositing. It could be in large*
*part but really it is just a hypothesis that needs further exploration.*

(AUTHORS)
We really thank the reviewer for his/her appreciation of our work.
We totally agree that regarding the possible sources of organic matter, we just reported some
hypotheses. With the available data-set in fact we were just able to make an hypothesis about the main
sources of DOM in the different sampling periods. The sampling periods cover a generally wide time
interval, and the deposition data are the result of integrating deposition over air masses of different origin
and with different aerosol characteristics and loads. This makes very difficult to find a correlation between
aerosol origin and DOC input, since the different sources are mixed in our samples.
If we would have reduced the sampling periods, we would have had less variability in the sources,
but in most of them we would have not had enough DOM to do all the analysis. Two weeks was therefore
the best compromise we were able to find.
We added in the conclusions a sentence highlighting that in order to understand the link between
aerosol origin and DOM concentration and quality further studies are necessary.

*Also, I was surprised not to consider wind direction properties when analyzing deposited material.*
*Lampedusa is a small island but I would not be surprised that when wind blows from directions other than*

*due East, and especially when it blows over the island from the West, substantial OM could be picked up*

*from the island itself.*

We thank the reviewer for this comment. At first, we took into consideration the air masses trajectories, but then we realized that there are some limitations in the use of wind direction to infer the aerosol sources.

Firstly, as above reported, the sampling periods cover a generally wide time interval, and the deposition data are the result of integrating deposition over air masses of different origin and with different aerosol characteristics and loads. Thus, in some cases information on the wind might suggest what are the dominant (if existing) wind conditions during a specific period; but due to possible differences in aerosol amounts and deposition, these conditions may not be representative of the integrated samples.

Secondly, the use of wind direction and speed identify the air mass origin (which is not what the reviewer is suggesting) may be problematic, since trajectories arriving to Lampedusa may take different paths depending on the synoptic conditions. A more elaborated approach would be required (e.g., trajectory reconstruction using wind, as in Becagli et al., 2012; or modeled backward trajectories based on meteorological analyses, as in Marconi et al., 2014). However, also in this case, the relatively long duration of the sampling interval and the variability of the deposition would prevent a robust attribution of the source regions.

Wind measurements conversely, as correctly suggested by the reviewer, might potentially provide useful information on the impact of local sources. The main local source area of anthropogenic particles is in the sectors between South and South-East of the sampling site; in this sectors there are the Lampedusa town, the power plant, the airport and the port. Previous studies have shown that wind from these directions is relatively infrequent, and the impact of these sectors is estimated to be negligible (see e.g., Artuso et al., 2009, with respect to atmospheric $CO_2$ measurements; Calzolai et al., 2015, with respect to PM10 measurements). This impact is expected to be even lower over samples integrated over are relatively large number of days.

Due to these reasons, we have preferred to associate the deposition samples' characteristics with those of PM10 samples collected daily and over the same time period. This allows to use aerosol properties measured in the same time intervals and influenced by the same sources to infer some overall conditions. Previous studies (e.g., Becagli et al., 2012, 2013, 2017; Marconi et al., 2014; Calzolai et al, 2016) have been dedicated at linking the PM10 measured composition with different aerosol sources.

*A third aspect of the manuscript deals with estimating the local and Mediterranean basin-wide importance of such deposition estimates for the biogeochemical functioning of the Mediterranean. I like this part myself but I have to admit it is the least elaborated since it is based on assumptions that will be hardly met. For instance, calculations based on the extension to the whole Mediterranean of the measured*

*OM deposition at Lampedusa. Given it is so variable and without a clear reason, I would expect variability to increase when other locations are taken into account. Also, the lability of the deposited organic matter is an unknown, so the final role of the marine biota is also unknown. But anyhow, I like these exercises. Thus, to me the main value of the manuscript is to provide a much needed data series of OM deposition measurements.*

We totally agree with the reviewer and we are aware that reporting the calculations based on the extension of our results to the entire Med Sea is a risk, because it is nor simple, nor appropriate, to assume that what is observed at one location is valid for the entire basin. However, we consider this as a first conceptual exercise that uses the new results from our study to give an estimate of the implications of DOM deposition for marine ecosystem, that needs to be supported by additional data.

To the best of the author's knowledge only one paper reports that a not-well quantified fraction of atmospheric DOM can be recalcitrant. Due to the lack of information, we decided to discuss implications taking into consideration both the possibilities: DOM is labile and DOM is recalcitrant. In the revised manuscript we will better stress the need for further investigations about the biological lability of DOM coming from the atmosphere

In order to better stress that these calculations are a conceptual exercise, we added the following sentences in the revised manuscript:

*"A conceptual exercise can be made in order to give an estimate of the implications of DOM deposition for marine ecosystem."*

*"Even if we are aware that these assumptions are hardly meet, in particular the estimate of DOC input to the whole Med Sea, based on the data collected in Lampedusa, we think that these calculations can give an idea of the relevant role that atmospheric input of DOC can have in sustaining bacterial productivity in the surface layer, particularly when the upper water column is strongly stratified."*

*The methods are standard within the field and thus assure quality control. Maybe I am not clear whether monthly data were calculated and how or whether just sample data was provided always? or in what cases? That is, how where data treated when more than 1 sample per month was available? How was the data split when covering periods from two consecutive months?, etc.*

We apologize for the inaccuracy. We did not calculate monthly data, we reported the sample data. In the figures 2, 3, 4, 6 and 7, the width of the bars refers to the duration of the sampling period (Table 1). This aspect will be clarified in the revised manuscript.

As a general rule, samples were collected every ~15 days, or immediately after strong rain or dust storm events. However, due to logistic problems the sampling period was longer than 20 days for 9 depositions (Table 1).

The DOC, DON and DOP fluxes, reported in the text and in the figures 2, 3, 4, 6 and 7 were calculated using the following formula:

                      $$X_{Flux} = X \cdot V/A \cdot d$$

  where X is the concentration of DOC, DON or DOP measured in the sample and expressed in $\mu M$; V

is the volume of rain collected by the sampler (expressed in L) or the volume of Milli-Q water used to wash the funnel walls in case of dry deposition (250 ml); A is the area of the funnel ($0.1018$ m$^2$), and d refers to the number of days of the sampling period. The DOC, DON and DOP fluxes are reported in the figures considering the flux corresponding to each sampling period. A paragraph with this explanation was added in the materials and methods section of the revised manuscript in order to clarify these calculations.

  *I understand that sample data is clearly reported in Fig. 5, but how were the rest treated is a bit*

*mysterious, especially since bars have unequal width within and between figures.*

  The bars in the figure 5 corresponded to the C:N:P molar ratios (see Table 3), so they referred to a number, not a flux, and this is the reason why the width of the bars is always the same in figure 5, in contrast to fig. 2, 3, 4, 6 and 7, where the width of the bars is different since it refers to the length of sampling periods. We added the description of the bars in the caption.

  *In line 150 it is also important to know the flow rate of the low-volume sampler. Also, I guess that*

*because of physical flow rate constraints a 1 μm filter could not be used. That would have been much*

*more desirable since there tend to be organic rich particles at the very fine particle ranges, and they would*

*have been missed, not a minor issue in this paper on OM. I would like the authors to comment on the*

*choice of a 2 μm filter to collect particles.*

  The filters used in this study are those usually used for aerosol sampling, they have a nominal porosity of 2 $\mu m$, but they are certified for 99% efficiency for particles having 0.3 $\mu m$ diameter.

  The sampling flow is maintained constant at 2.3 m$^3$/h in order to maintain constant the sampling heads cut-off (10$\mu m$) as reported in the European rule UNI EN12341.

  In order to clarify these concepts for a broad number of readers the text was changed as follows:

   *"PM10 (particulate matter with aerodynamic equivalent diameter lower than 10 μm) is routinely sampled on*

*a daily basis at the island of Lampedusa (Becagli et al., 2013; Marconi et al., 2014; Calzolai et al., 2015) by using a*

*low-volume  dual-channel sequential sampler (HYDRA FAI Instruments) equipped with two PM10 sampling heads,*

*operating at constant flow of 2.3 m3/h in accord with the European rules for aerosol monitoring (UNI EN12341).*

*Aerosol is collected on 47 mm diameter Teflon filters (PALL Gelman) having 2 μm nominal porosity but certified to*

*have 99% retention efficiency for 0.3 μm diameter particles. The PM10 mass was determined by weighting the Teflon*

*filters before and after sampling with an analytical balance in controlled conditions of temperature (20±1 °C) and*

*relative humidity (50±5%)."*

*The manuscript is well structured and balanced. The title is informative of the contents. The language is proficient. Figures should be uniformed or clarified in aspects such as the x-axis but are otherwise well done. Other than that, I have no major concerns publishing the manuscript pretty much as it is.*

We really thank the reviewer for his/her appreciation of our manuscript. We reworked the figures in order to uniform them and to eliminate any misunderstanding.

**Reply to Referee 2**

*(REFEREE)*

*This paper presents a sound dataset concerning the dry and wet deposition fluxes of dissolved organic matter sampled for 2 years and a half at the island of Lampedusa (Italy). This site, in the central Mediterranean, is appropriately taken to represent the interaction atmosphere-sea surface in a remote marine environment. It is a well written paper which addresses a topic of interest: the role of DOM (and its components DON and DOP) deposition in the western Mediterranean. It explores the role of the frequent Saharan intrusions, a very interesting point since few studies have dealt with the interactions between organic carbon and Saharan dust. Finally, it specifically addresses the role of this atmospheric deposition for marine productivity.*

*The quantification of N and P atmospheric deposition to the Mediterranean has been previously addressed in many papers, the most relevant of them are adequately cited by the authors. However, I'd like to bring to the authors attention the work of Izquierdo et al. 2012 in Atmospheric Environment. Atmospheric phosphorus deposition in a near-coastal rural site in the NE Iberian Peninsula and its role in marine productivity, since it will provide more data for comparison, discussion and understanding of the role of African sources in marine biogeochemistry, and the relative contribution of dry and wet deposition.*

*The layout of the paper and data treatment are OK, and I have only a few suggestions, which I list below.*

(AUTHORS)

We really thank the reviewer for his/her appreciation of our manuscript. In the revised version, all
the comments and suggestions were taken into consideration. In the revised manuscript, the work of
Izquierdo et al. (2012) was cited as suggested by the reviewer.

*Introduction*
*Lines 34-35. Industrial pollution can also be originated from North Africa as has been shown in*
*the work of Rodríguez et al (2011). Transport of desert dust mixed with North African industrial*
*pollutants in the subtropical Saharan Air Layer. Atmospheric Chemistry and Physics 11, 6663–6685. I*
*think it is worth considering.*

In the introduction of the revised manuscript, we added this reference and a sentence about the
possible contribution of pollution from North Africa: "Industrial pollution can also be originated from the North
Africa as shown in the work by Rodríguez et al. (2011)."

*Line 40. The work of Izquierdo et al 2012 could be included in this list of references, since it deals*
*with how P dep influences the marine biogeochemical cycle in the western Med Sea.*

In the introduction of the revised manuscript, we added this reference.

*Line 45. This sentence should be revised as it is not true that atmospheric deposition affects*
*radiative forcing and human health. Aerosols in the atmosphere do, but not deposition.*

We agree with the reviewer that the sentence was not clear. In the revised manuscript, we changed it
as follows: "*Atmospheric deposition of organic carbon can therefore affect regional C cycling (Yan and*
*Kim, 2012; Decina et al., 2018).*"

*Material and methods*
*I recommend to make some reorganization of the text, since some paragraphs in this section in*
*fact correspond better to the Introduction. E.g. the paragraph dealing with the explanation of the*
*Mediterranean seawater DOM stoichiometry compared to the world oceans (lines 70-75) should be moved*
*to the Introduction.*

We apologize for this inaccuracy, in the revised manuscript, this sentences was deleted.

*Same thing with the paragraph justifying the appropriateness of Lampedusa as representing an*
*unpolluted site in the central Med.*

In agreement with this comment, in the revised manuscript the paragraph 2.1 was changed, reducing
the excessive descriptive part.

*Line 78. Revise the notation of units of mean dust deposition*
OK.

*Line 95. polycarbonate, not in capital letter Paragraph*
OK.

*104-108. Please list in this text the ions and metals analyzed*
We listed the ions and metals in section 2.5 and we deleted this part since it is also reported at lines
135-140, as noted by the reviewer.

*Line 105 and 136. blank levels, instead of blanks level*
OK.

*Lines 135-140. This has been already exposed in lines 104-108.*

As above reported, we deleted the lines 104-108 and we added here the list of metals and ions.

*Line 144. I see that the particulates retained in the filters (after wet and dry deposition filtration)*
*was analysed. But the procedure of digestion and analysis is not reported. Same thing for particulates*
*from the PM10 samples (line157). This should be described in the M&M.*

In agreement with this comment, in the revised manuscript, the description of the procedure were
added in the Material and Methods section.

*Results*
*Line 202. Here there is an error, since the upper limit of TDP is 5\*10 exp-3 (as deduced from*
*Table 2).*

We apologize for the inaccuracy. The mistake was corrected in the revised manuscript.

***Line 244. Error in unit: 8.8 ug m-3***

We apologize for the inaccuracy. The mistake was corrected in the revised manuscript.

***Discussion***

***In this section I'd like a more in deep discussion of dry versus wet deposition and its relation to***
***meteorology.***

In agreement with this comment, in the revised manuscript, we added some information on dry
versus wet deposition and its relation to meteorology. In particular, in the results we reported the annual
rainfall during 2016 and in the description of the study area we added the following sentence: "*Precipitation*
*shows a significant interannual variability and is concentrated in autumn and winter, with a maximum in*
*October. Intense precipitation events, which are relatively infrequent, are generally associated with frontal*
*passages and winds from the Northern sectors. Very dry conditions characterize late spring and summer.*" In
the discussion (paragraph 4.2), we added the following sentence : "*All of the analyzed samples, except few*
*cases in summer 2016, are relative to dry+wet conditions. Although the DON and DOP recorded during the*
*dry samplings are generally on the low end side of the measured range (see Table 2), no information on the*
*role played by wet and dry deposition processes may be drawn at this stage, due to the limited number of dry*
*samples.*" Regarding DOC input, in the discussion (paragraph 4.1) we reported that "*Finally, the correlation*
*between monthly precipitation rates and DOC fluxes shows the importance of rain events as a source of*
*DOC in the Med Sea, as proposed by Djaoudi et al. (2018).*"

In the literature, the wet atmospheric deposition is considered the main pathway for the removal of
organic carbon from the atmosphere. While our results that dry deposition is also important and we have
reported a detailed discussion of this point in the paragraph 4.3: "Some models have estimated that wet
deposition represents up to 75-95% of total deposition (Iavorivska et al., 2016). Our data While confirm the importance
of wet deposition, it also stress the relevance of dry deposition (32% of the total deposition during the entire sampling
period) that appears to be the main contributor of DOC and of other chemical species to the remote site of Lampedusa,
as suggested by Morales-Baquero et al. (2013)."

***High DOC deposition was recorded in Lmp25 (May 2016) and also in Lmp1 (end of March 2015)***
***and Lmp 4 (May2015) coinciding with Saharan dust but low DOC was found in Saharan events during***
***autumn and winter. In view of this clear seasonal differentiation, one could hypothesize that there is a***
***role of pollen attached to desert particles in these spring events ( end March-May) and this pollen would***
***contribute DOC. This process would not occur in the other seasons (winter and autumn of no pollen***
***production). This is a possible explanation that needs further attention. However, there are some reports***

*in the literature of joint pollen and dust transport: for example, Van Campo and Quet (1982) identified pollen types transported from North Africa to south France together with mineral desert dust, Franzen et al. 1994 documented the arrival of pollen from the Mediterranean to Fennoscandia during a dust event. Pollen originating in Morocco was detected South Spain (Cabezudo et al. 1997) and various pollen types (Cannabis, Cupressus, Pinus, Platanus and Sambucus) were observed in Cordoba (South Spain) exclusively during dust African events (Cariñanos et al. 2004).*

We really thank the reviewer for this interesting suggestion. The contribution of pollen to atmospheric DOC in spring is an interesting hypothesis to test. In the revised manuscript, the following sentence was added in the discussion (paragraph 4.3): "*In addition Lmp01 (end of March 2015), Lmp04 (May 2015) and Lmp25 (May 2016) show a seasonality that could be linked to the transport of pollen attached to desert particles in the spring events, and this pollen would contribute to atmospheric DOC input in spring (end of March- May). Pollen originating in Morocco was detected in South Spain (Cabezudo et al., 1997) and various pollen types (Cannabis, Cupressus, Pinus, Platanus and Sambucus) were observed in Cordoba (South Spain) exclusively during dust African events (Cariñanos et al., 2004). This process would not occur in the other seasons (winter and autumn), when no pollen production occurs.*" A sentence about the need of further studies was added in the conclusions.

*Figures In fig 2, 3 and 4, include a legend to indicate the color of wet and dry deposition.*

In the revised manuscript, a legend was included in figures 2, 3 and 4.